# Analysis of nearly 3000 archaeal genomes from terrestrial geothermal springs sheds light on interconnected biogeochemical processes

Yan-Ling Qi[1,8], Ya-Ting Chen[2,8], Yuan-Guo Xie[1], Yu-Xian Li[1], Yang-Zhi Rao[1], Meng-Meng Li[3], Qi-Jun Xie[1], Xing-Ru Cao[1], Lei Chen[1], Yan-Ni Qu[1], Zhen-Xuan Yuan[1], Zhi-Chao Xiao [1], Lu Lu[4], Jian-Yu Jiao[3], Wen-Sheng Shu [5], Wen-Jun Li [3] ✉, Brian P. Hedlund [6,7] ✉ & Zheng-Shuang Hua [1] ✉

Terrestrial geothermal springs are physicochemically diverse and host abundant populations of Archaea. However, the diversity, functionality, and geological influences of these Archaea are not well understood. Here we explore the genomic diversity of Archaea in 152 metagenomes from 48 geothermal springs in Tengchong, China, collected from 2016 to 2021. Our dataset is comprised of 2949 archaeal metagenome-assembled genomes spanning 12 phyla and 392 newly identified species, which increases the known species diversity of Archaea by ~48.6%. The structures and potential functions of the archaeal communities are strongly influenced by temperature and pH, with high-temperature acidic and alkaline springs favoring archaeal abundance over Bacteria. Genome-resolved metagenomics and metatranscriptomics provide insights into the potential ecological niches of these Archaea and their potential roles in carbon, sulfur, nitrogen, and hydrogen metabolism. Furthermore, our findings illustrate the interplay of competition and cooperation among Archaea in biogeochemical cycles, possibly arising from overlapping functional niches and metabolic handoffs. Taken together, our study expands the genomic diversity of Archaea inhabiting geothermal springs and provides a foundation for more incisive study of biogeochemical processes mediated by Archaea in geothermal ecosystems.

As the least-understood domain of the tree of life, the Archaea is a major frontier of microbiology and biogeochemistry research[1,2]. Known Archaea shoulder key roles in biogeochemical cycles[3] by mediating unique or unusual metabolisms such as methanogenesis,

anaerobic methane oxidation, anaerobic alkanotrophy, and chemolithotrophic ammonia oxidation[4,5]. Although some groups of Archaea can be readily cultivated in the lab, less than 0.1% of the known species diversity of Archaea has been described as pure cultures[6].

[1]Chinese Academy of Sciences Key Laboratory of Urban Pollutant Conversion, Department of Environmental Science and Engineering, University of Science and Technology of China, Hefei 230026, China. [2]Institute for Disaster Management and Reconstruction, Sichuan University–Hong Kong Polytechnic University, Chengdu 610207, China. [3]State Key Laboratory of Biocontrol, School of Life Sciences, Sun Yat-Sen University, Guangzhou 510275, PR China. [4]College of Environmental Science and Engineering, China West Normal University, Nanchong 637009, China. [5]School of Life Sciences, South China Normal University, Guangzhou, PR China. [6]School of Life Sciences, University of Nevada Las Vegas, Las Vegas, NV 89154, USA. [7]Nevada Institute of Personalized Medicine, University of Nevada Las Vegas, Las Vegas, NV 89154, USA. [8]These authors contributed equally: Yan-Ling Qi, Ya-Ting Chen. ✉e-mail: liwenjun3@mail.sysu.edu.cn; brian.hedlund@unlv.edu; hzhengsh@ustc.edu.cn

The explosion of archaeal metagenome-assembled genomes (MAGs) in recent decades has greatly expanded our scope of view of the true diversity of Archaea, but still more than half of archaeal taxonomic 'blind spots' have not yet been described carefully in the literature and await 'discovery'[7].

Due to their physicochemical extremes—especially temperature and pH - geothermal ecosystems are rich in extremophilic Archaea, while more moderate habitats within geothermal areas are conducive to less extremophilic Archaea. Prokaryotes and their viruses dominate in high-temperature geothermal ecosystems because no known eukaryotes can grow above ~65 °C; and within these systems, Archaea are known to dominate at the high-temperature and low pH extremes, since no known Bacteria can grow above 95 °C[8] or below pH 2[9,10]. Physicochemical extremes are very common in terrestrial geothermal ecosystems because subsurface phase separation and varying degrees of mixing with shallow water can lead to wide variations in spring temperature and chemistry[11]. In these systems, acidic pools represent vapor-dominated end-members along with sulfur-depositing fumaroles; the low pH of these systems derives from the enrichment of hydrogen sulfide in the vapor phase, followed by oxidation of the hydrogen sulfide to sulfuric acid[12]. Alkaline springs represent water-dominated end-members and are characterized by high content of total dissolved solids, while high-temperature, high-pH spring sources are significantly understudied[11,13].

One magmatic hydrothermal region that harbors springs with such physicochemical extremes is the Tengchong geothermal spring system in Southwest China, which belongs to the Mediterranean-Himalayan Geothermal Belt that resulted from collision of the Indian Plate and the Eurasian Plate. Pronounced variations regarding the physicochemical parameters have been observed among different geothermal springs, with reported temperatures up to 96 °C and pH values from ≤1.8 to ≥9.3[11,14,15]. Despite decades of microbiology

research at Tengchong geothermal springs, very few archaeal isolates have been described, and all belong to the genera *Sulfolobus*, *Acidianus*, and *Metallosphaera*[16–19], within the order *Sulfolobales*. Recently, metagenomics has uncovered the potential metabolisms of some yet-uncultivated archaeal lineages in Tengchong geothermal springs, including chemolithotrophic *Caldarchaeales*[20,21], symbiotic *Aenigmatarchaeota*[22] and *Panguiarchaeales*[23], carbohydrate-metabolizing *Bathyarchaeia*[24], ammonia-oxidizing *Nitrosocaldaceae*[25], anaerobic methylotrophic class EX4484-205[26], and several lineages potentially involved in methane/alkane metabolism[27].

However, while these studies provided important insights into specific lineages of Archaea, they did not provide a high-level view into the full diversity and potential functionality of Archaea as a whole, or the forces controlling their diversity and distribution. Here, we applied deep metagenomic and metatranscriptomic sequencing to diverse sediment samples from a variety of geothermal springs in Tengchong. A total of 152 metagenomes were acquired from 48 springs over a period of 6 years, and nearly 3000 archaeal MAGs were successfully reconstructed. By analyzing these MAGs and focusing on highly transcribed genes from metatranscriptomes, we describe the diversity, distribution, and potential roles of Archaea in the biogeochemical cycling of carbon, sulfur, nitrogen, and hydrogen. Our study expands the global cache of archaeal genomes and illustrates their potential ecological roles and interactions in terrestrial geothermal ecosystems.

## Results
### Nearly 3000 archaeal genomes adequately represent the Archaea diversity in Tengchong geothermal springs
A total of 152 sediment samples covering 48 geothermal pools/streams located in Tengchong County (Yunnan Province, China) and spanning six years from 2016 to 2021 were collected and metagenomically sequenced (Supplementary Data 1). A comprehensive investigation of

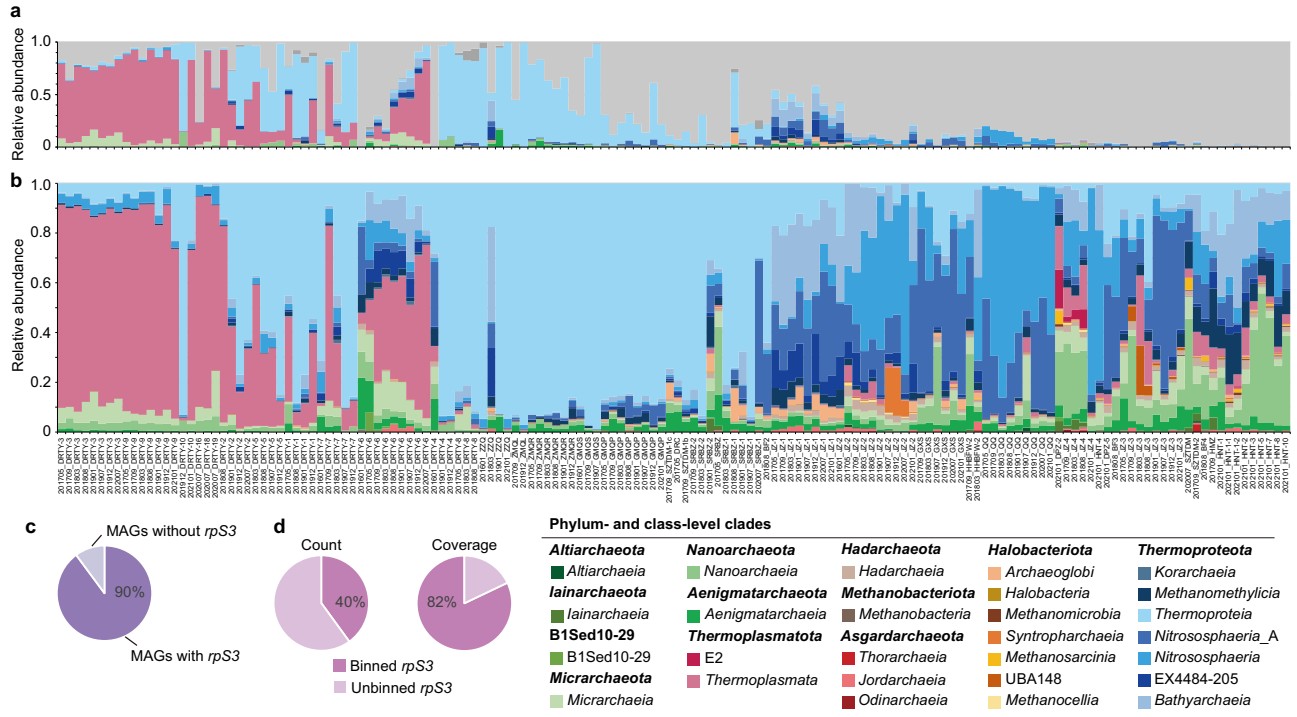

**Fig. 1 | Archaeal diversity in 152 geothermal spring metagenomes.**
**a** Composition and relative abundance of archaeal classes in microbial community based on *rpS3* gene sequences. Detailed information including the relative abundance of each *rpS3* gene in each microbial community is reported in Supplementary Data 2. **b** Composition and relative abundance of archaeal classes in archaeal community based on rMAGs. Colors indicate archaeal classes: green series, DPANN superphylum; blue series, *Thermoproteota*; red series, *Asgardarchaeota*. **c** The percent of archaeal MAGs harboring *rpS3* genes among all 2949 archaeal MAGs. **d** The percent of *rpS3* genes associated with archaeal MAGs normalized to the total archaeal community based on count and coverage of *rpS3* genes.

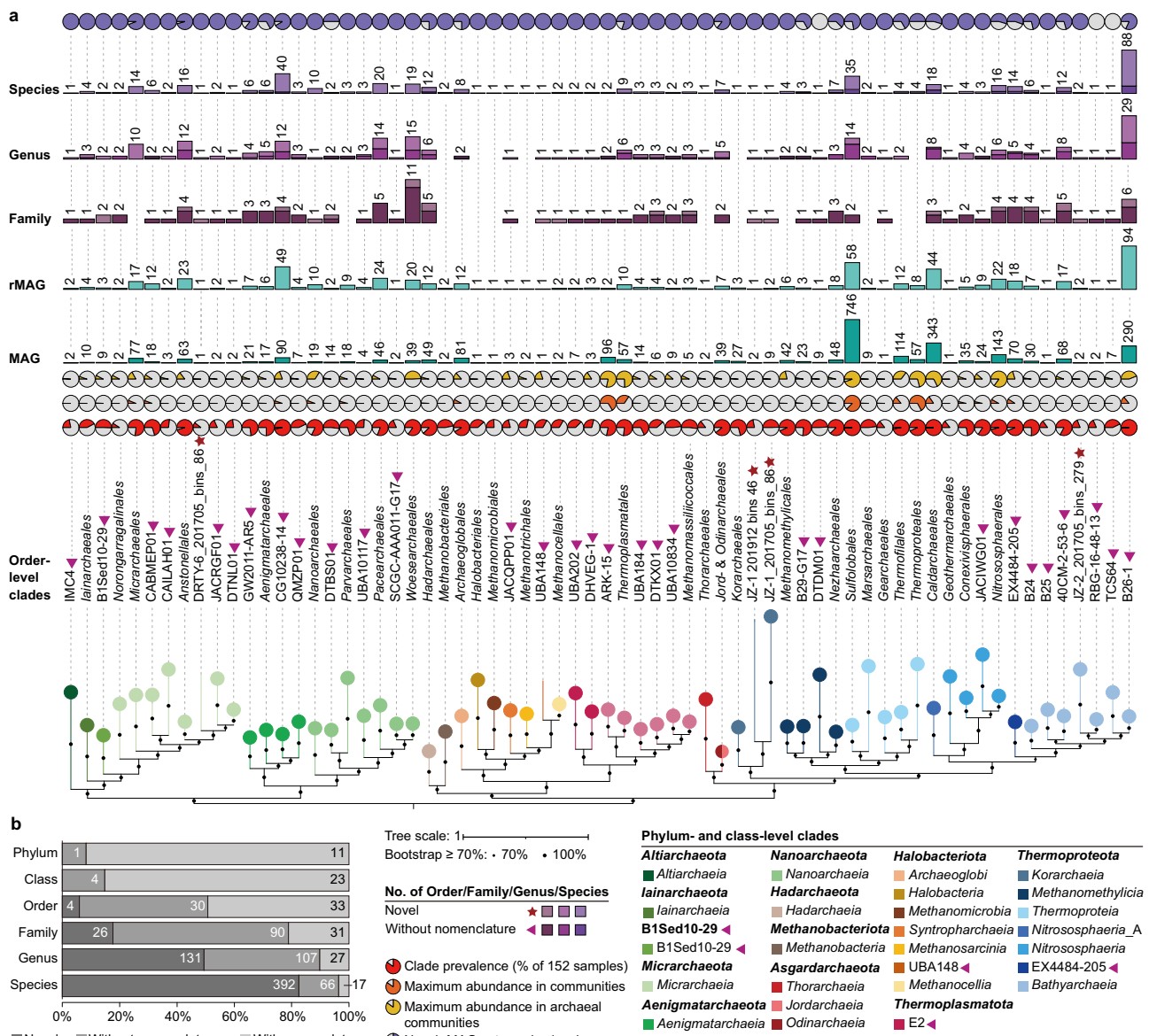

**Fig. 2 | Newly identified archaeal species are broadly distributed across the Archaea. a** Phylogenetic tree of 603 archaeal rMAGs recovered from Tengchong geothermal springs. Taxonomic labels are based on the GTDB. Triangles behind taxonomic labels indicate the order/class existed in GTDB but without nomenclature; stars behind taxonomic labels indicate the order is newly discovered in this study. Red pie charts indicate the prevalence of each order across 152 samples. Orange and yellow pie charts indicate the maximum relative abundance of each order in microbial and archaeal communities, respectively. Dark blue and light blue columns indicate the numbers of archaeal MAGs and rMAGs contained in the corresponding order, respectively. Purple stacked histograms indicate the number of families, genera, and species that are newly identified (light part) and existing but without nomenclature (dark part). Purple pie charts indicate the percentage of newly identified species among all species in each order. The relative abundance of each order in microbial communities is available in Supplementary Data 6. The generated tree in newick format is provided in FigShare repository at https://doi.org/10.6084/m9.figshare.25650441. **b** Number and percentage of taxa with newly identified and existing but not formally named at each taxonomic level for 603 archaeal rMAGs.

archaeal diversity was conducted by initially scanning for genes encoding ribosomal protein S3 (*rpS3*). Using this procedure, we successfully identified 27,436 archaeal and bacterial *rpS3* sequences, which clustered into 8757 representative *rpS3* sequences with 99% as the sequence identity cutoff (see Methods). Taxonomic assignments based on the Genome Taxonomy Database (GTDB) revealed that archaeal *rpS3* genes could be classified into 13 phyla, 29 classes, and 64 orders (Supplementary Fig. 1 and Supplementary Data 2). Further bioinformatic analysis resulted in the construction of 2949 archaeal MAGs meeting currently accepted criteria as medium to high quality[28], with a median of 89.6% completeness and 0.8% contamination (Supplementary Fig. 2, Supplementary Data 3 and 4). Moreover, these MAGs had a median of 70 scaffolds, and over 90% (2,678) of the MAGs

had ≤200 scaffolds. The quantity of our reconstructed MAGs is equivalent to about half of the existing archaeal genomes in the GTDB (total 6062 Archaea genomes in GTDB v.207 until Jan. 2023)[29]. Although approximately 10% of the archaeal MAGs did not contain *rpS3* (Fig. 1c), the other 90% of the MAGs represent 40% of the total archaeal representative *rpS3* by number, and importantly represent 82% of the archaeal representative *rpS3* when they were weighted by sequence coverage in the metagenomes (Fig. 1d). Also, according to the GTDB nomenclature, these MAGs encompass 12 of the 13 phyla and 25 of the 29 classes detected, and the taxonomy of the MAGs is highly consistent with that derived by the representative *rpS3* (Fig. 1b, Supplementary Fig. 1, Supplementary Data 5 and 6). Overall, these results show that the 2,949 archaeal MAGs recovered in our study adequately

represent the archaeal community in the Tengchong geothermal spring system.

As expected, the composition of Archaea varied greatly across samples (Fig. 1a, b and Supplementary Fig. 1). The average and maximum abundance of Archaea based on reads mapping to MAGs was 34.7% and 95.2%, respectively, and 54 samples were dominated by Archaea (>50% relative abundance; Supplementary Data 6). This is further supported by qPCR experiments conducted on several recent samples collected from geothermal spring sediments (Supplementary Table 1). In agreement with many previous thermal spring surveys[30,31], we found that *Thermoplasmata* and *Thermoproteia* were the most abundant lineages across our sites (Fig. 1a, b). *Thermoplasmata* dominated (maximum abundance, 70.2%) the DRTY pools, which are acidic (pH <5.5), followed by *Thermoproteia*, *Nitrososphaeria*, and *Bathyarchaeia*, as well as *Micrarchaeia* and *Nanoarchaeia*, which has recently been validated as *Nanobdellia*[32] (Fig. 1b, Supplementary Data 1 and 6). However, *Thermoproteia* dominated in most other samples, including both acidic springs (some DRTY, ZZQ pools) and alkaline springs (pH ≥ 8.5; including ZMQR/L, SRBZ, GMQS/P pools), with a maximum abundance of 93.7%. The composition of other Archaea in *Thermoproteia*-dominant communities in acidic springs was similar to those in *Thermoplasmata*-dominated acidic springs, whereas in alkaline springs other Archaea included *Nitrososphaeria*_A, *Aenigmatarchaeia*, *Methanomethylicia*, *Nanoarchaeia* (*Nanobdellia*), and *Archaeoglobi*. The remaining springs (such as JZ, GXS, QQ, HNT pools) contained various Archaea that together contributed up to 45% of the read abundance. Given the high abundance of Archaea in the Tengchong spring system, we further investigated their diversity and metabolic potential.

## MAGs are diverse and represent hundreds of unknown species

To better explore the diversity of these Archaea, the 2949 archaeal MAGs were dereplicated at the strain level into 603 representative MAGs (rMAGs) based on 99% average nucleotide identity (Supplementary Data 3). The rMAGs represent the majority of archaeal phyla (12 of 18, 66.7%), as defined in the GTDB, and 27 classes, 67 orders, 147 families, 265 genera, and 475 species (Fig. 2). They comprise the first genomic representatives of four orders (within *Micrarchaeia*, *Korarchaeia*, and *Bathyarchaeia*), 26 families, and 131 genera. Additionally, they include one phylum (B1Sed10-29), three classes (belonging to *Halobacteriota*, *Thermoplasmatota*, *Thermoproteota*), 30 orders, 90 families, and 107 genera that defined in GTDB without nomenclature and effective descriptions. Of the 475 species represented by the rMAGs, 82.5% do not have a previous species representative in GTDB, and another 13.9% do not have a nomenclature. Thus, the Archaea in Tengchong geothermal springs are poorly explored and they are of great research value.

The identified rMAGs are taxonomically broadly distributed (Fig. 2). In particular, they span a broad diversity within the *Thermoproteota* (7/8 classes) and several phyla known for their small cell size, small genome size, and episymbiotic lifestyles (*Altiarchaeota*, *Iainarchaeota*, B1Sed10-29, *Micrarchaeota*, *Aenigmatarchaeota* and *Nanoarchaeota* (*Nanobdellota*)), with hotspots of diversity and novelty in the orders B26-1 (*Bathyarchaeia*), *Sulfolobales*, *Caldarchaeales*, and CG10238-14 (within *Aenigmatarchaeia*). Further, many *Thermoproteota* orders were both prevalent and abundant across the 152 samples, especially *Sulfolobales*, *Thermofilales*, *Thermoproteales*, *Caldarchaeales*, *Nitrososphaerales*, and B26-1, suggesting their ecological importance. Small Archaea known or interpreted to be episymbionts were widely distributed, but at low abundance. Only *Micrarchaeales*, CABMEP01 (within *Micrarchaeia*), CG10238-14, *Nanoarchaeales* (*Nanobdelalles*), and *Parvarchaeales* had relative abundances greater than 1% in at least one metagenome (Supplementary Data 6). By contrast, fewer than 10 rMAGs were assigned to orders within *Thermoplasmatota*, *Hadarchaeota*, *Halobacteriota*, *Methanobacteriota*, and

*Asgardarchaeota*, except for the more diverse orders *Archaeoglobales*, *Thermoplasmatales*, and *Hadarchaeales*. It is noteworthy that ARK-15 (*Thermoplasmata* order), with only two rMAGs, was abundant (up to 66%) in some of the metagenomes. In addition to abundant species, these rMAGs also represent diverse low-abundance species with several representatives, further confirming this collection of genomes represents the diversity of the Archaea community well. In summary, the identified MAGs in geothermal springs not only improve genomic representatives within several archaeal lineages, but also substantially increase the known archaeal phylogenetic diversity across the tree of life.

## pH and temperature are key drivers of archaeal composition and potential function

By applying a generalized linear model (GLM), we found that microbial communities correlated with a variety of environmental factors (Supplementary Table 2). Among them, pH and temperature emerged as the two strongest correlates and they most likely affect the archaeal community at both species ($x^2$ = 40.8 and 29.2, respectively, both $p < 0.001$) and functional ($x^2$ = 46 and 34.2, respectively, both $p < 0.001$) levels. Sulfate and nitrite were weaker correlates at both species ($x^2$ = 8.42 and 3.84, respectively, both $p < 0.05$) and functional ($x^2$ = 8.18 and 4.00, respectively, both $p < 0.05$) composition of Archaea. Our analyses of both taxonomic composition and potential metabolic functions showed remarkable variation according to pH (Supplementary Fig. 3a-d). Statistical differences among MAGs present in acidic (pH <5.5), neutral (5.5 ≤ pH <8.5) and alkaline (pH ≥ 8.5)[33] springs were evident (ANOSIM test, $R = 0.807$ and $p = 0.001$ for rMAGs; $R = 0.638$ and $p = 0.001$ for KOs). Additionally, grouping samples by temperature revealed significant differences among mesothermal (<60 °C), thermal (60–80 °C), and hyperthermal (≥80 °C)[34] communities (ANOSIM test, $R = 0.275$ and $p = 0.001$ for rMAGs; $R = 0.255$ and $p = 0.001$ for KOs; Supplementary Fig. 3f, h), suggesting temperature is a secondary factor that also influences the taxonomic and functional diversity of geothermal spring archaeal communities. The composition and potential function of Archaea from thermal communities was similar to those from mesothermal communities but vastly different from those hyperthermal springs based on both the ordination and correlation coefficient (Supplementary Fig. 3e–h). There was no significant difference in archaeal communities when grouped by sampling season, despite previous reports of seasonal effects on chemistry and community composition in Tengchong springs[35] (Supplementary Fig. 3i–l).

## Archaea are more abundant under extreme pH conditions and show a strongly bimodal distribution

Archaeal community composition varied significantly between acidic, neutral, and alkaline springs. Additionally, correlation analyses reveled strong relationships between the relative abundance of Archaea and pH (Supplementary Fig. 4a, b). When the springs were divided into acidic or alkaline groups, both extremes of pH favored archaeal abundance, with stronger relationships in acidic springs (Pearson test, $R = -0.76$, $p = 5.6 \times 10^{-14}$) than alkaline springs (Pearson test, $R = 0.57$, $p = 2.7 \times 10^{-7}$). Quantitative data obtained through qPCR also indicated that in acidic springs, the absolute abundances of Archaea was greater than those of Bacteria (Supplementary Table 1). This trend is consistent with the known competitive advantage of Archaea over Bacteria at several physicochemical extremes[36]. However, in alkaline springs, both pH (Pearson test, $R = 0.55$, $p = 5.5 \times 10^{-7}$) and archaeal abundance (Pearson test, $R = 0.67$, $p = 1.1 \times 10^{-10}$) correlated with temperature (Supplementary Fig. 4c–f). Further, analysis of the relationship between pH and abundance of the 20 most abundant archaeal orders showed *Thermoproteia* and *Thermoplasmata* dominating, but with distinct patterning by pH (Supplementary Fig. 5). *Sulfolobales* dominated in

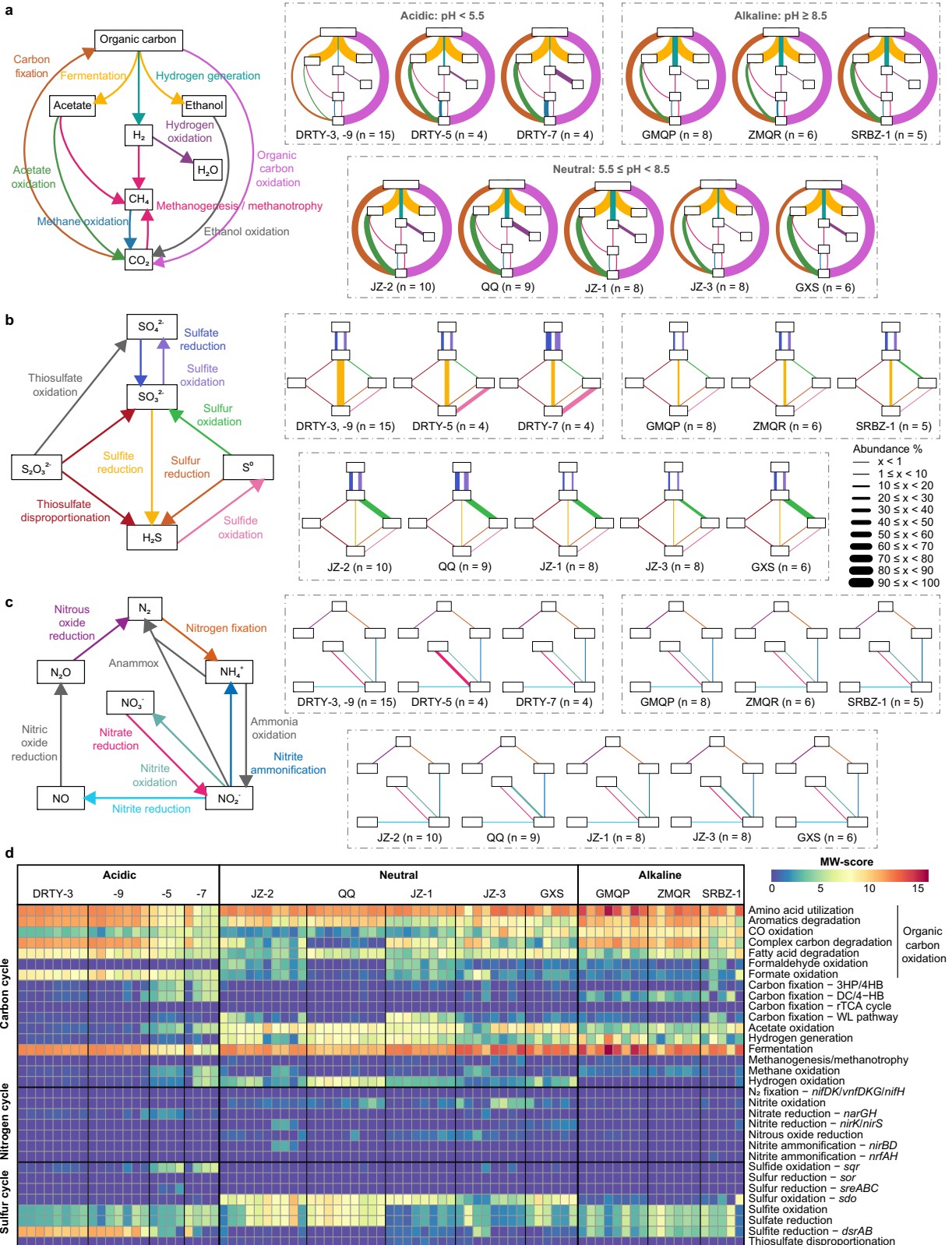

**Fig. 3 | Metabolic profile of archaeal communities under different pH conditions.** Schematic diagrams of the carbon and hydrogen cycle (**a**), sulfur cycle (**b**), and nitrogen cycle (**c**) within the archaeal communities in geothermal springs. Legends for each cycle are shown on the left; gray lines indicate pathways that were not detected. The panels on the right indicate the reactions encoded by the archaeal communities in the different springs, and the thickness of the line indicates the relative abundance of Archaea putatively involved in the reaction. The geothermal springs are grouped by pH: the three on the upper left are acidic, the three on the upper right are alkaline, and the five on the lower are neutral. Only the geothermal springs that were sampled ≥ 4 times are shown, and the values of relative abundance represent the average. **d** Metabolic weight score (MW-score; see Methods) of archaeal communities in different geothermal springs.

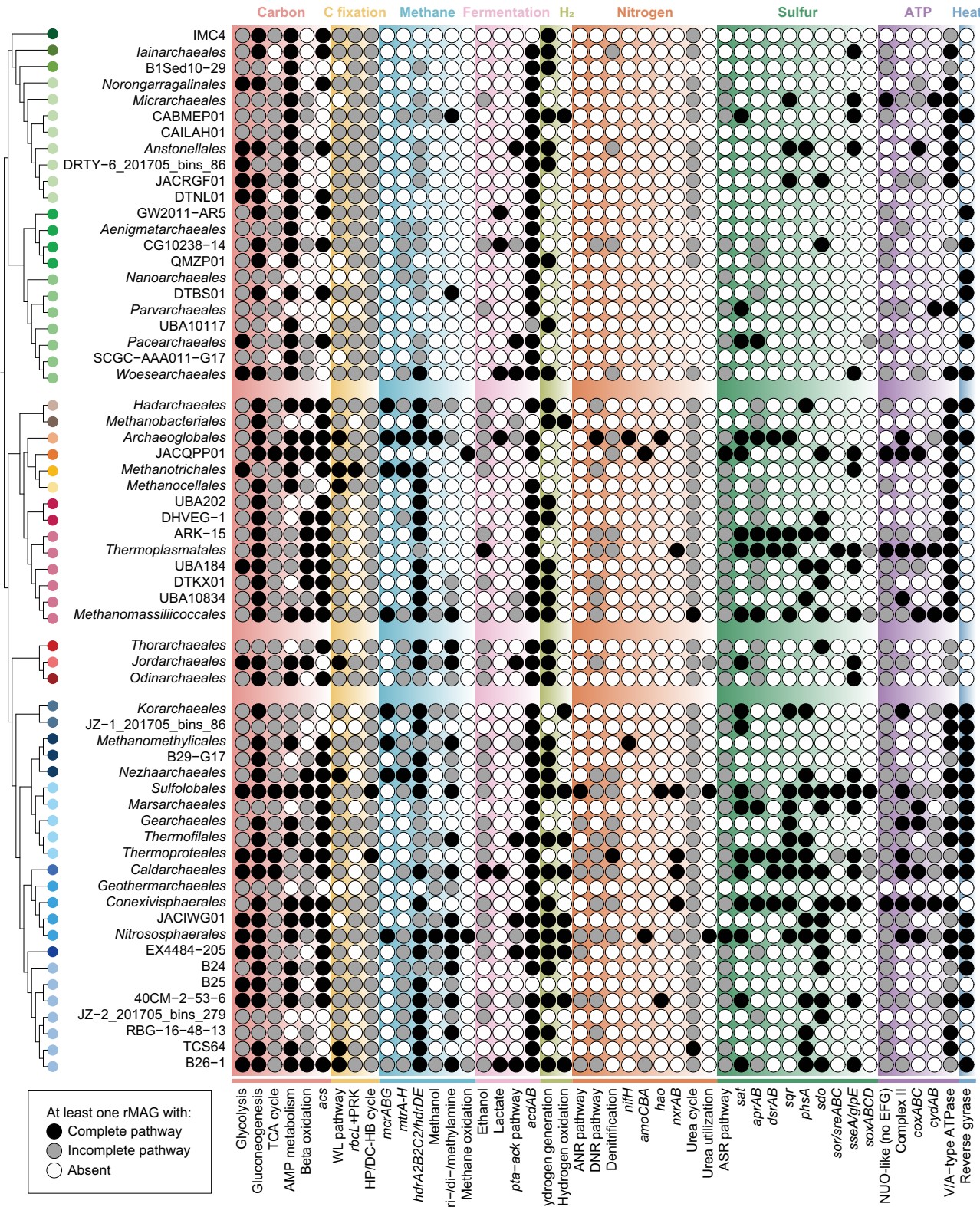

**Fig. 4 | Metabolic potential of archaeal orders in the geothermal spring metagenomes.** Black circles indicate the existence of a complete pathway, that is, all genes in the pathway are present. Gray circles indicate the existence of an incomplete pathway. White circles indicate the absence of any genes in the pathway. For each archaeal order, pathways containing all genes in at least one rMAG are considered representative of the order. The counts of functional genes for each MAG are summarized in Supplementary Data 7.

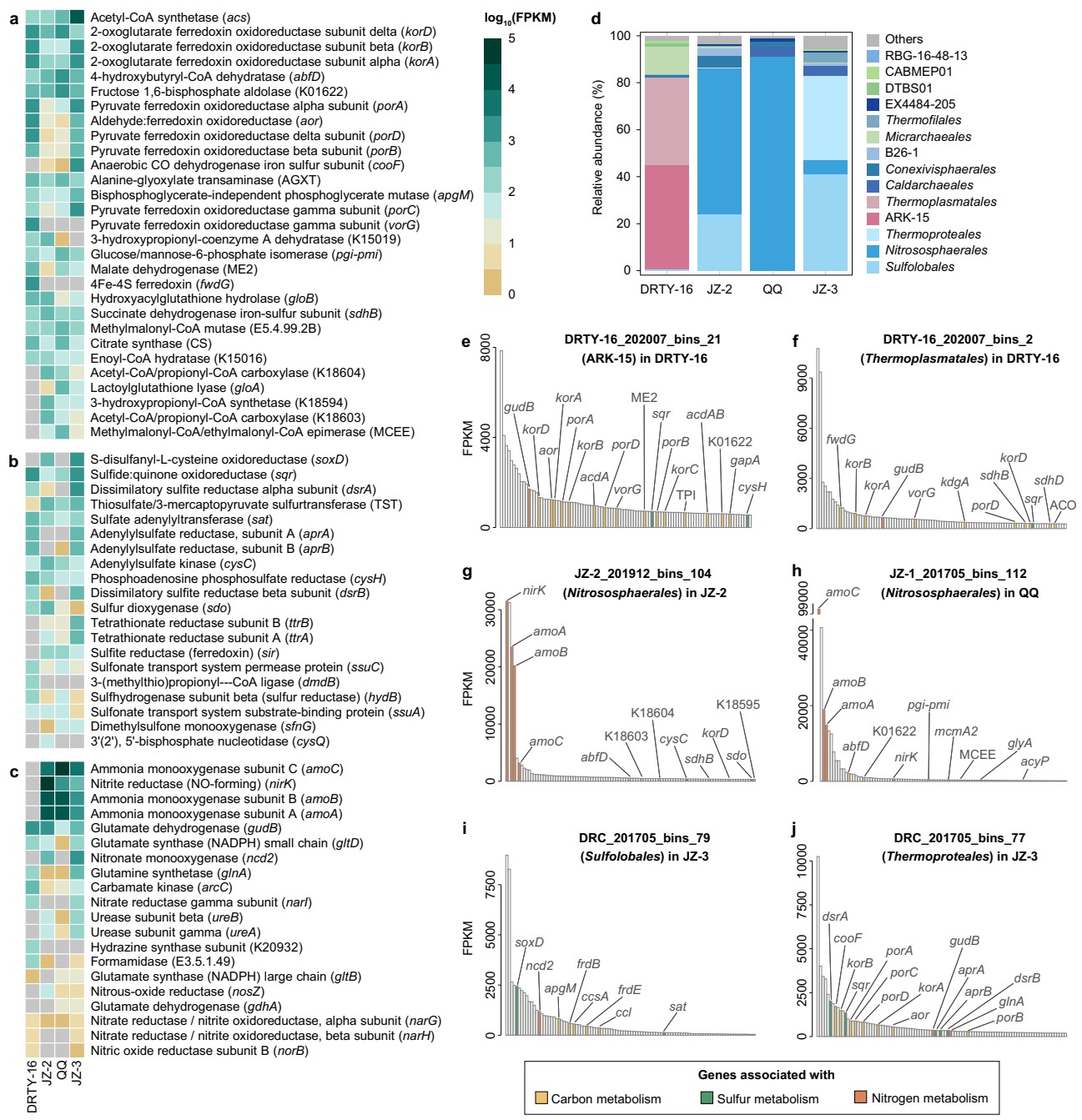

**Fig. 5 | Metatranscriptomic read mapping to archaeal MAGs in geothermal springs.** Top 10 highest expressed genes associated with carbon metabolism (**a**), sulfur metabolism (**b**), and nitrogen metabolism (**c**) in the four reference springs. Cells in gray indicate either the archaeal genomes contained no ortholog or no expression of the ortholog is detected. **d** Percent relative abundance of meta-transcriptomic reads mapping to the top five archaeal orders in each reference spring. The top 100 transcriptional rank of functional genes of rMAGs from the most abundant orders in the reference springs, including DRTY-16_202007_bins_21 (ARK-15; **e**) and DRTY-16_202007_bins_2 (*Thermoplasmatales*; **f**) in DRTY-16 pool, JZ-2_201912_bins_104 (*Nitrososphaerales*; **g**) in JZ-2 pool, JZ-1_201705_bins_112 (*Nitrososphaerales*; **h**) in QQ pool, DRC_201705_bins_79 (*Sulfolobales*; **i**) and DRC_201705_bins_77 (*Thermoproteales*; **j**) in JZ-3 pool. FPKM: Fragments Per Kilobase of transcript per Million.

many acidic springs. For example, *Sulfolobaceae* dominated in DRTY-4, −5, −10 and ZZQ pools (Supplementary Fig. 5–6 and Supplementary Data 6), consistent with its abundance in other acidic thermal springs[37,38]. *Sulfolobales* were also abundant in some alkaline springs. For example, *Desulfurococcaceae* were abundant in ZMQL/R and GMQP pools, with a maximum relative abundance of 87%. Two *Thermoproteia* orders, *Thermoproteales* and *Thermofilales*, were more abundant in alkaline springs, as well as *Nezhaarchaeales*. In contrast, two orders of *Thermoplasmata*, ARK-15 and *Thermoplasmatales*, were enriched in acidic springs, with maximum abundance of 66% and 34%, as well as other acidophilic lineages like *Micrarchaeia*, *Parvarchaeales*, JACIWG01, and *Conexivisphaerales*. Several lineages with lower abundance were more abundant in neutral springs, such as *Caldarchaeales*, B26-1, *Nitrososphaerales*, and EX4484-205, whose maximum abundances are 15%, 14%, 8% and 9%. We also observed that some widespread small Archaea, including *Micrarchaeales*, *Parvarchaeales*, and CABMEP01, existed exclusively in acidic springs, most likely because of their symbiotic/parasitic

relationships with acidophilic hosts[39–41] (Supplementary Fig. 7). Overall, the archaeal communities showed a selective enrichment in response to pH.

### *Thermoproteota* are likely key players in the carbon cycle

An analysis of potential carbon metabolisms encoded by rMAGs within 12 representative springs that were sampled more than four times was conducted to investigate metabolic potential (Fig. 3). Given the strong relationships between pH and potential archaeal functions (Supplementary Fig. 3c, d), the pattern of potential autotrophic pathways according to pH was examined. In neutral springs, the Wood-Ljungdahl pathway dominated, and B26-1 was the largest archaeal contributor (Figs. 3a, d, 4). *Thermoproteales* encoding a complete set of genes involved in the dicarboxylate-hydroxybutyrate (DC/4-HB) cycle including the key enzymes acetyl-CoA/propionyl-CoA carboxylase (K01964, K15036, K15037) were abundant in both acidic and alkaline springs (Supplementary Data 7). Additionally, in acidic springs, members of *Sulfolobales* were abundant and encoded either the 3-hydroxypropionate/4-hydroxybutyrate (3HP/4HB; with pyruvate synthase and PEP carboxylase being the key enzymes (K01007, K01595)) or DC/4-HB cycles, as supported by the detection of a complete set of genes for both pathways. Surprisingly, some members of the *Sulfolobales* were highly abundant in alkaline springs but only possessed the 3HP/4HB cycle, suggesting that 3HP/4HB might also be an important pathway for carbon fixation in such environments. Genes encoding potential for methane metabolism, notably methyl-coenzyme M reductase (*mcrABG*), were restricted to low-abundance orders, such as putative methanogens within the *Archaeoglobales*, *Methanotrichales*, *Methanomassiliicoccales*, *Nezhaarchaeales*, *Korarchaeales*, *Methanomethylicales*, *Nitrososphaerales* and putative methanotrophs/alkanotrophs in the *Hadarchaeales* (Fig. 4), which were described in detail previously[27]. Even though some members of the *Archaeoglobales* and *Nezhaarchaeales* in Tengchong and other terrestrial spring systems have recently been shown to be methanogenic[42,43], the number of transcripts mapping to *mcrABG* in our datasets were low.

Acetate is a prevalent electron donor and carbon source in both oxic and anoxic environments, and species-specific uptake of acetate was recently studied in a Tengchong spring[44]. Furthermore, acetate oxidation with low potential electron acceptors, such as partially oxidized sulfur compounds, can be an important intersection of the carbon cycle and other element cycles[45]. Genes for acetate oxidation were observed in most springs (average 8–60% coverage except DRTY-3 and DRTY-9; Fig. 3a). Particularly, the acetyl-CoA synthetase (*acs*) gene was detected among many archaeal orders, likely functioning in acetate oxidation to acetyl-CoA (Fig. 4 and Supplementary Data 7). The importance of acetate oxidation was further supported by metatranscriptomic sequencing, with reads mapping to *acs* in many springs (Fig. 5a). In addition to acetate oxidation, archaeal genes related to fermentation were abundant across all representative springs (over 53% coverage) further confirming the presence of a baseline population of heterotrophic Archaea (Fig. 3a). The highly expressed pyruvate:ferredoxin oxidoreductase (*por*) and related poorly characterized tungsten-dependent aldehyde ferredoxin oxidoreductases (*aor*) are important for both sugar and amino acid fermentations and were recently shown to have proliferated in the *Caldarchaeales* family *Wolframiiraptoraceae* in Tengchong and other geothermal ecosystems[21] (Fig. 5a). Ferredoxins Fer primarily facilitating the transfer of electrons from glucose and/or amino acids to oxidoreductases were commonly detected across multiple orders within *Thermoproteota*[46] (Supplementary Data 7). Additionally, the widespread existence and expression of acetate-CoA ligases (*acdAB*; Fig. 4 and Supplementary Data 8) indicates that many Archaea can oxidize acetyl-CoA for energy conservation. However, when carbon sources become limited or acetate becomes the sole carbon source, these Archaea can utilize the reverse ACD and POR enzymes to assimilate

acetate as a carbon substrate[45,47,48]. Interestingly, another acetogenesis pathway, the phosphate acetyltransferase (*pta*)-acetate kinase (*ack*) pathway, was also detected in small Archaea (e.g., *Pacearchaeales* and *Woesearchaeales*), *Asgardarchaeota*, and *Thermoproteota*. The *pta-ack* pathway is thought to be a bacterial-type acetogenesis pathway, although it has been found in a few Archaea[49,50]. Our discovery of widespread *pta-ack* pathway enzymes suggests that this so-called Bacteria-specific pathway is more widely distributed in Archaea than previously thought.

The dominance of rMAGs coding for organic carbon metabolism (over 92% coverage) across all sites suggests the presence of a "baseline" of heterotrophic Archaea (Fig. 3a). However, genes related to heterotrophic processes were distinct in the three pH groups, likely indicating preferences for distinct organic carbon substrates (Fig. 3d), mirroring differences in dissolved organic carbon pools in acidic and alkaline springs in Tengchong and elsewhere[51]. For example, Archaea in neutral springs tended to encode more families of carbohydrate-active enzymes (CAZymes) than those in acidic springs, even though they contained fewer CAZymes per rMAG (Supplementary Fig. 8). Those in alkaline springs harbored a similar number of CAZymes families as those in neutral springs, but they had more CAZymes per rMAG. Archaea in acidic springs encoded fewer CAZymes per rMAG and fewer CAZyme families, and polysaccharide lyases were nearly absent. Although many acidic springs are known to accumulate terrestrial organic carbon[51], these patterns suggest that polysaccharides may be acid-hydrolyzed naturally in acidic springs, rendering CAZymes less important. We note that order EX4484-205, also known as *Brockarchaeota*, is a major contributor to CAZymes in the circumneutral springs, consistent with previous reports[26] (Supplementary Fig. 9), suggesting that EX4484-205 may play an important role in degrading diverse complex organics in neutral springs. Similarly, several orders of *Thermoproteia* contained high numbers of CAZymes in both acidic and alkaline springs. *Sulfolobales* CAZymes associated with the metabolism of amino sugars and *Thermoproteales*/*Thermofilales* CAZymes associated with starch/hemicellulose degradation were both highly expressed in the alkaline JZ-3 spring (Supplementary Data 9). Additionally, in acidic DRTY-16 spring, starch-degrading CAZymes encoded by *Thermoplasmatales* and ARK-15 were expressed. The potential carbon metabolisms encoded by archaeal communities in acidic springs showed two distinct patterns (Fig. 3d). In some pools, notably DRTY-3 and DRTY-9 (DRTY-3/-9), the heterotrophic "baseline" dominated, while in others, notably DRTY-5 and DRTY-7 (DRTY-5/-7), Archaea encoded those functions plus carbon fixation and acetate oxidation. This difference correlated with the high abundance of *Thermoproteia* in DRTY-5/-7 (Fig. 1), suggesting these *Thermoproteia* may be important for those functions.

### Potential for dissimilatory sulfate reduction is enriched in polyextreme geothermal springs mainly by *Thermoplasmata* in acidic springs and *Thermoproteales* in alkaline springs

For Archaea to thrive in extreme habitats, it would be opportune to diversify to exploit diverse energy sources. Genes supporting sulfite oxidation and sulfate reduction were ubiquitous (Fig. 3b, d), and the key gene sulfate adenylyltransferase (*sat*) was expressed in all four metatranscriptomic datasets (Fig. 5b). Both metagenomic and metatranscriptomic analyses showed the prevalence of the dissimilatory sulfite reductase (*dsrAB*) and adenylylsulfate reductase (*aprAB*) gene clusters across both acidic and alkaline springs, suggesting the importance of dissimilatory sulfate reduction (DSR) (Figs. 3b, 5b), although very few measurements of DSR in high-temperature terrestrial geothermal ecosystems have been reported[52]. In acidic geothermal springs, DSR was likely mediated by the most abundant Archaea, ARK-15, whereas the high expression of both *dsrAB* and *aprAB* in some *Thermoproteales* suggested distinct sulfate-reducing Archaea in neutral and alkaline springs (Fig. 5j). Notably, the most abundant ARK-15

rMAG lacks *aprAB* genes, indicating a potential dependency on partners to provide sulfite. The sulfide derived from DSR or from the spring sources themselves[12] could be oxidized using sulfide:quinone oxidoreductase (*sqr*) to form zero-valent sulfur, a crucial intermediate in the biogeochemical sulfur cycle that is widespread in terrestrial and marine geothermal ecosystems[53]. In acidic springs, many rMAGs within the orders ARK-15 and *Thermoplasmatales* possessed these genes and their transcripts were also abundant; likewise, many rMAGs encoding these genes belonged to the *Thermoproteales* in alkaline springs (Fig. 5 and Supplementary Data 8). In contrast to these dissimilatory pathways in the polyextreme springs, Archaea in neutral springs expressed the suite of genes necessary for assimilatory sulfate reduction (Fig. 5b). Additionally, the broad occurrence of sulfur dioxygenases (*sdo*) suggested chemolithotrophic sulfur oxidation in neutral springs, especially by B26-1, *Caldarchaeales*, *Nitrososphaerales*, and EX4484-205 (Figs. 3d, 4). These *sdo* genes were also highly expressed in neutral JZ-2 pool and predominantly map to *Nitrososphaerales* (Supplementary Data 8). We also noted a high number of transcriptional reads mapping to thiosulfate sulfurtransferase (*sseA*) for thiosulfate oxidation to form sulfite in neutral and alkaline pools, and tetrathionate reductase (*ttrAB*), for reduction of tetrathionate to thiosulfate, in alkaline JZ-3 pool. In summary, both metagenomic and metatranscriptomic data suggested an active dissimilatory sulfur cycle mediated by Archaea in acidic pools, which may be driven by the concentration of sulfur in the vapor phase initially as hydrogen sulfide. The data suggest that hydrogen sulfide can be oxidized by Archaea, and the resulting sulfate is used by DSR. In alkaline springs, the data suggest that Archaea can oxidize reduced sulfur and then utilize various sulfur compounds including tetrathionate, thiosulfate, sulfate, and sulfite to conduct DSR or form zero-valent sulfur.

An interesting and unique case is found in the DRTY-6 pool. The pH of DRTY-6 pool gradually decreased through time from neutral to acidic, accompanied by an increased abundance of Archaea, among which the class *Thermoplasmata* was dominant (Fig. 1a and Supplementary Fig. 10). Meanwhile, we observed a gradual increase in DSR pathway genes. This transformation reinforces the link between *Thermoplasmata* and the DSR pathway as a response to the enrichment of sulfate in acidic springs. Surprisingly, the abundance of archaeal genes for sulfur oxidation decreased as DRTY-6 acidified through time. We also noticed a gradual change in potential carbon metabolisms among the archaeal community; for example, the abundance of genes for inorganic carbon fixation, acetate oxidation, and hydrogen metabolism decreased. This is consistent with a decline in the relative abundance of *Thermoproteia*. Combining these trends with decreases in carbohydrate and hydrogen metabolism, we surmise that the acidification of DRTY-6 may have led to a shift in the predominant archaeal pathways for energy conservation, with more genes related to DSR rather than sulfur oxidation.

## Multiple strategies for utilization of nitrogen resources in N-limited geothermal springs

According to the GLM results mentioned above, nitrite significantly correlated with the archaeal species and functional composition in geothermal springs (Supplementary Table 2). However, the abundance of genes for dissimilatory nitrogen metabolisms in Archaea in Tengchong springs were limited relative to carbon and sulfur, which is consistent with previous studies[24,54] (Fig. 3c, d). The concentrations of dissolved inorganic nitrogen are low in most neutral and alkaline springs[35]. Nitrogen is supplied to many terrestrial geothermal springs as ammonia[55], which can be oxidized to nitrite by thermophilic ammonia-oxidizing Archaea (AOA)[56,57]. We detected a high number of transcriptional reads mapping to ammonia monooxygenase (*amoCAB*), suggesting active ammonia oxidation by AOA (Fig. 5 and Supplementary Data 8). Both *amoCAB* and nitrite reductase (*nirK*) were highly expressed by MAGs assigned to *Nitrososphaerales* inhabiting neutral and alkaline springs. The high expression of *nirK* by *Nitrososphaerales* suggests that the greenhouse gas $N_2O$ may be released by nitrifier denitrification (Fig. 5c). Ammonia oxidation to nitrite or nitrate could provide substrates for assimilatory or dissimilatory metabolisms. The archaeal communities in these springs encode more genes associated with dissimilatory nitrate reduction (DNR) than assimilatory nitrate reduction (Fig. 4), suggesting AOA provide sufficient oxidized nitrogen to drive denitrification, although in other geothermal ecosystems, the rate of denitrification is limited by the nitrate/nitrite supply[56]. Despite the expression of genes involved in the DNR pathway in DRTY-16, the complete lack of expression of *amoCAB* suggests that the dominance of ammonium over ammonia may inhibit ammonia oxidation at low pH (Fig. 5c, Supplementary Data 8). Community-level metabolic reconstructions revealed that incomplete denitrification may be an important archaeal pathway particularly in neutral springs (Fig. 3c, d).

In addition, Archaea living in geothermal springs may obtain nitrogen from organic molecules. ARK-15 rMAGs expressed formamidase (E3.5.1.49) in acidic springs, which suggests transformation of methanamide to ammonia (Fig. 5c and Supplementary Data 8). Some *Sulfolobales* (*Desulfurococcaceae* members) in neutral and alkaline springs expressed nitronate monooxygenase (*ncd2*), which converts nitroalkane into nitrite. Urease (*ureAB*) transcripts mapping to *Nitrososphaerales* suggests they may produce ammonia from urea in JZ-2 pool. Surprisingly, *ureAB* were also highly expressed in alkaline JZ-3 pool by members of the *Jordarchaeales*, suggesting broad use of urea in geothermal ecosystems. Moreover, the expression of glutamate dehydrogenase (*gudB*), which interconverts glutamate and ammonia, was much higher than glutamine synthetases (*glnA*, *gltBD*) that couple ammonium assimilation to glutamate synthesis (Fig. 5c). *gudB* was encoded by ARK-15/*Thermoplasmatales* and *Thermoproteales* from acidic DRTY-16 and alkaline JZ-3 pools, respectively (Fig. 5e, f, j).

## Archaeal hydrogen metabolism is widely distributed in geothermal springs

Molecular hydrogen occurs ubiquitously in geothermal springs from both geological and biological sources, and it is a widely utilized electron donor and intercellular electron shuttle for microbial growth and survival[58]. The potential metabolism of hydrogen by the archaeal community in geothermal springs was different under different pH conditions (Fig. 3a, d). Hydrogenases were identified in nearly half of the archaeal rMAGs (270 out of 603, 45%; Supplementary Data 10), including [NiFe]-, [FeFe]- and [Fe]-hydrogenases (Supplementary Fig. 11). The $H_2$-oxidizing [Fe]-hydrogenases, which are specific to *Methanobacteriales*[4], although *mcrABG* were not detected in the *Methanobacteriales* rMAGs from this study. Several Group A [FeFe]-hydrogenases, known to be involved in coupling ferredoxin oxidation to fermentative $H_2$ evolution, were detected in *Micrarchaeia* and *Nanoarchaeia* (*Nanobdellia*), as described from single-amplified genomes of *Iainarchaeota* previously[59]. However, [FeFe]- and [Fe]-hydrogenases were far less abundant compared with [NiFe]-hydrogenases.

[NiFe]-hydrogenases were the most widespread hydrogenases, occurring in 11 phyla, 22 classes, and 38 orders of Archaea (Supplementary Data 10 and Supplementary Fig. 11). This pattern mirrors the overall diversity of [NiFe]-hydrogenases, which are divided into four clades: $H_2$-uptake Group 1 (membrane-bound) and 2 (cytosolic), bidirectional Group 3, and $H_2$-evolving Group 4[59,60]. Group 1a,g,j [NiFe]-hydrogenases were distributed in a few orders of *Thermoproteota*. These enzymes are $O_2$-sensitive and mediate anaerobic $H_2$-dependent sulfate, nitrate, metal, heterodisulfide, and fumarate respiration[60]. Group 2e [NiFe]-hydrogenases are a unique lineage of aerobic uptake hydrogenases and were only found in aerobic hydrogenotrophic *Thermoproteia*[61]. As expected, a Group 2e [NiFe]-hydrogenase was encoded by a *Sulfolobales* rMAG (DRTY-4_201912_bins_2) that was highly abundant in many acidic springs. Group 3 [NiFe]-hydrogenases

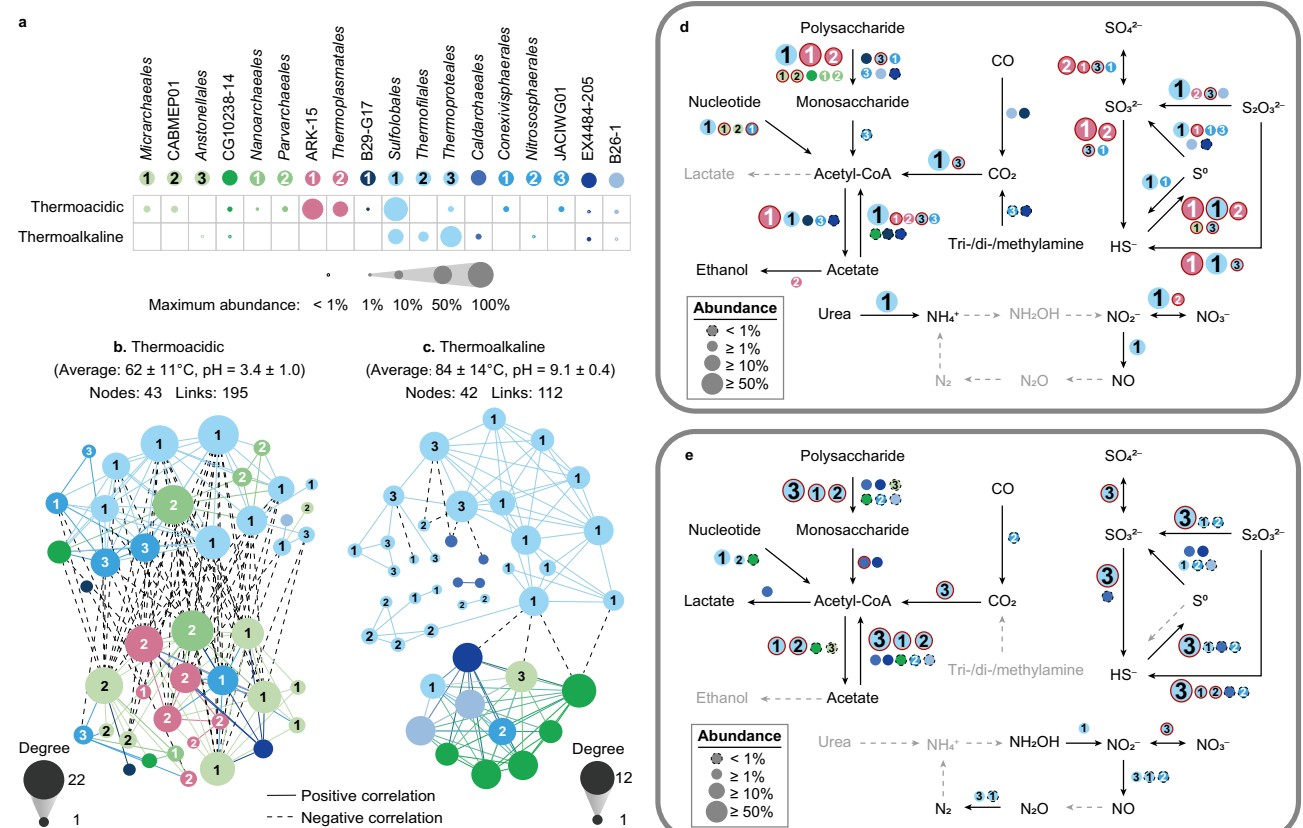

**Fig. 6 | Potential metabolic interactions within archaeal communities in Archaea-dominant geothermal springs. a** Maximum relative abundance of archaeal orders involved in co-occurrence networks in acidic (pH <5.5) and alkaline (pH ≥8.5) geothermal springs. The co-occurrence networks between archaeal orders in thermoacidic (**b**) and thermoalkaline (**c**) springs. Schematic diagram of distinct archaeal orders involved in co-occurrence networks and their potential roles in the biogeochemical cycles of carbon, nitrogen, and sulfur metabolism in acidic (**d**) and alkaline (**e**) geothermal spring habitats. The color of circle indicates class, which is consistent with Fig. 1. The number in circle indicates the order assigned to same class.

are cytosolic bidirectional enzymes coupling with different cofactors: $F_{420}$ (3a), NADP (3b), heterodisulfide reductase (3c), and NAD (3d). Group 3b and 3d [NiFe]-hydrogenases were encoded by the neutral-abundant B26-1, EX4484-205, *Nitrososphaerales*, and *Caldarchaeales*; these are $O_2$-tolerant and are considered as redox valves that interconvert electrons between $H_2$ and NAD(P)H depending on the availability of exogenous electron acceptors[62]. These Group 3b and 3d [NiFe]-hydrogenases may help these Archaea to cope with oxygen fluctuations. Group 3c [NiFe]-hydrogenases form functional complexes with heterodisulphide reductase (MvhADG-HdrABC) and perform the electron bifurcation process that couple the endergonic reduction of ferredoxin with $H_2$ by coupling it with the exergonic reduction of heterodisulfide with $H_2$[63]. This hydrogenase was prevalent in *Halobacteriota*, *Thermoplasmatota*, and *Thermoproteota*. Moreover, the Archaea rMAG dataset encoded diverse membrane-bound $H_2$-evolving Group 4 [NiFe]-hydrogenases, with several subgroups associated with different electron donors, namely formate (4b) and ferredoxin (4d, 4e, 4 g, 4 h, 4i). These enzymes have a respiratory function that releases electrons from low-potential donors to reduce protons, preserving the free energy liberated during electron transfer as a proton- or sodium-motive force[63]. The majority of Group 4 [NiFe]-hydrogenases were assigned to subgroup 4b and 4 g and were prevalent in members of *Sulfolobales* and *Thermofilales* enriched in alkaline springs, along with B26-1 in neutral springs. Additionally, ferredoxins involved in hydrogen production by facilitating electron transfer to Group 4 [NiFe]-hydrogenases were also commonly observed among Archaea[46] (Supplementary Data 7). Altogether, the diverse hydrogenases likely sustain $H_2$-based respirations,

fermentations, and various electron-bifurcation and energy-conversion mechanisms in these archaeal communities.

## Some Archaea may be competitive or cooperative

We focus not only on the metabolic potential of individual archaeal orders, but also attempt to explore the larger biogeochemical processes and metabolic interactions, including competitive or cooperative dynamics within the archaeal community. As mentioned above, Archaea were not abundant in the relatively temperate springs (low-temperature neutral pH springs), where most ecosystem functions are likely performed by Bacteria (Fig. 1a). Therefore, we built co-occurrence networks based on communities dominated by Archaea (>50% in relative abundance) in polyextreme (thermoacidic and thermoalkaline) springs. The networks constructed based on the investigation of *rpS3* genes demonstrated that there are only few links between Archaea and Bacteria, and most of these links exhibited significant negative correlations. This suggested a potential competitive rather than cooperative relationship between Archaea and Bacteria (see microbial networks at https://doi.org/10.6084/m9.figshare.25650441). To enhance the identification of potential beneficial functional interactions among Archaea with greater resolution and precision, we extended our analysis to include archaeal rMAGs in constructing co-occurrence networks (Fig. 6a). In thermoacidic springs, the network exhibits high phylogenetic diversity, including 14 archaeal orders. The dominant lineage, *Thermoplasmata*, may compete with another dominant lineage, *Sulfolobales*, as illustrated by negative correlations between the two cohorts (Fig. 6b). This competition may be attributed to overlapping functional niches (Fig. 6d), and

the outcome of the competition is determined by temperature. When temperature exceeds 62 °C, *Sulfolobales* prevails over *Thermoplasmata* (Supplementary Fig. 12). Conversely, as the temperature decreases, *Thermoplasmata* gains a competitive advantage over *Sulfolobales*. Another factor, the high concentration of sulfate in DRTY pools, may further favor *Thermoplasmata*, especially the ARK-15 subgroup, members of which are capable of respiring sulfate or sulfite, providing an extra niche in the competition with other Archaea. Alternatively, some positive co-occurrence patterns suggest cooperativity (Fig. 6b, c). In thermoacidic springs, *Thermoplasmatales* and *Conexivisphaerales* may cooperate on sequential redox transformations of sulfur (Fig. 6b, d). *Thermoplasmatales* oxidize sulfide to elemental sulfur, which can be further oxidized into sulfite or reduced to sulfide by *Conexivisphaerales*. We conjecture that the partial DSR pathway of dominant ARK-15 species may require the reduction of sulfate by *Thermoplasmatales* and the low-abundance *Conexivisphaerales* for sulfite supply, though no significant relationship was observed between them. In contrast, fewer Archaea encode pathways for sulfate reduction in thermoalkaline springs, mainly members of the *Thermoproteales* and *Caldarchaeales* (Fig. 6c, e). Additionally, one module in the network in thermoalkaline springs consists of abundant *Sulfolobales* and *Thermoproteales*, while the other comprises less abundant Archaea, including small Archaea from CG10238-14 and *Anstonellales* (Fig. 6c). Although distinct, thermoacidic geothermal springs also host numerous low abundance small Archaea in the network, such as *Micrarchaeales*, CABMEP01, and *Parvarchaeales*, suggesting the importance of these poorly studied Archaea. Given the expression of CAZymes and *rbcL* gene in *Micrarchaeales* and CABMEP01, these less abundant small Archaea may participate in the degradation of complex organic carbon and nucleotides. This could potentially serve as a source of monosaccharides and phosphorus sources or as part of a parasitic or predatory relationship with their hosts such as ARK-15 and *Thermoplasmatales* (Fig. 6d, e; Supplementary Data 11). In general, the analysis of rMAGs supports the general notion that competition and cooperation between Archaea may be common, especially in springs where Archaea dominate over Bacteria.

## Discussion

We provide a large resource of archaeal genomes from a unique magmatic-geothermal spring area of mainland China that hosts physicochemically diverse springs. These 2,949 reconstructed genomes and over four million protein sequences considerably expand the known phylogenetic diversity of Archaea in terrestrial geothermal systems and expand the total number of archaeal genomes in the GTDB by over 48%. The integration of metagenomic and metatranscriptomic sequencing allowed us to examine the metabolic potential and gene expression and enabled us to develop conceptual models of potential metabolic interactions between Archaea at the ecosystem scale. Additionally, we provide evidence that microbially generated low pH not only controls community composition but by doing so it alters the abundance of functions for important biogeochemical cycling reactions. Within the dataset, temperature was also strongly correlated with archaeal diversity.

A fundamental goal of ecology is to understand how environmental parameters impact the composition and function of biological communities. For a long time, temperature and pH have been reported as the two most influential factors affecting the distribution of microbial communities in thermal springs, where temperature is usually considered to be the main driving factor. The general belief is that both microbial richness and diversity show a unimodal relationship with temperature, with diversity peaking around 20 to 40 °C[64–67]. However, many samples in these studies have a narrow pH range of 6–8, so pH effects may be underestimated. In contrast, several other studies with a wider range of pH suggest microbially generated low pH is the dominant correlate with the distribution and diversity of

microbial communities in thermal springs[68–70]. Springs in those studies are divided into two distinct clusters, acidic (2–5) and neutral (6–8) pH. There are very few high-temperature alkaline springs with pH above 8 in these studies. The largest data set of 925 hot springs studied by Power et al. also showed that temperature only has a significant effect above 70 °C[70]. Those results are similar to our study, which showed that archaeal composition and function are primarily correlated with pH at temperature below 80 °C (Supplementary Fig. 3). Unlike previous studies based on 16 S rRNA gene sequencing, the current study avoids primer bias and assumptions about function based on single gene amplicons. Additionally, our large data set spans a very wide range of pH (2–10) and temperature (23–100 °C) and is a good representation of the physicochemical diversity within terrestrial geothermal ecosystems worldwide. Our study shows that neither pH nor temperature alone is sufficient to explain the community composition, but emphasizes the importance of pH, especially in acidic springs buffered by sulfuric acid, as the strongest control.

Most -omics studies of Archaea focus on the dominant lineages, while low-abundance lineages are often overlooked, especially groups of small Archaea such as *Nanoarchaeota* (*Nanobdellota*), *Aenigmatarchaeota*, and *Micrarchaeota*, as well as *Asgardarchaeota*. Our metatranscriptomic analysis showed that some small Archaea, such as CABMEP01 (within *Micrarchaeia*) and CG10238-14 (within *Aenigmatarchaeia*), expressed multiple carbohydrate-metabolizing enzymes, peptidases, and carbohydrate and amino acid transporters (Supplementary Data 11). Furthermore, they encoded a number of transposases with high expression, suggesting frequent gene exchange, which may be relevant to their symbiotic lifestyle[71]. In addition, *Micrarchaeia* was the only member in acidic DRTY-6 spring expressing ribulose-bisphosphate carboxylase (*rbcL*), which is involved in the AMP pathway catabolizing nucleotides as reported previously[72] (Supplementary Data 8). Interestingly, members of the *Jordarchaeales*, an *Asgardarchaeota* order, were the only archaeal lineages that expressed urease and a urea transport system in JZ-3 pool (Supplementary Data 11). These enzymes may help *Jordarchaeales* utilize exogenous urea and convert it into $CO_2$ and ammonia, which has never been reported in *Jordarchaeales*, which generally have poor nitrogen-metabolizing capacity[73]. In short, low-abundance small Archaea and *Asgardarchaeota* occupy unique ecological niches in these geothermal ecosystems. Although methanogens were not abundant in these springs, microcosm experiments demonstrated that they were active but temperature-sensitive[41]. That study noted a shift in the predominant methanogenic pathway in a member of the *Archaeoglobales* from hydrogenotrophic to methylotrophic when incubation temperatures increased from 65 to 75 °C. And both genomic inference and *mcr* expression under methanogenic conditions strongly suggested hydrogenotrophic methanogenesis by a member of the *Nezhaarchaeales* in situ.

We note that some key biochemical functions are encoded by coexisting but taxonomically distinct orders of Archaea (Fig. 6). This functional redundancy is widespread in microbial communities in various environments[74–77]. In our geothermal springs, a high functional redundancy was observed with respect to energy-transducing metabolic pathways[78], including polysaccharide degradation, acetogenesis, acetate oxidation, thiosulfate disproportionation, and sulfide oxidation, which are conducted by abundant but phylogenetically diverse Archaea (Fig. 6). These functions are likely to be fundamental and important archaeal community processes in geothermal ecosystems[79], and support polysaccharide, acetate and sulfur compound as important energy source. In addition, in our dataset, functional redundancy seemed to be more prevalent in polyextreme conditions. An example is the distribution of CAZymes, which were more abundant and widespread in the archaeal community in acidic/alkaline springs relative to neutral springs within the same functional cluster (Supplementary Fig. 7). The identity of Archaea possessing the same function can vary substantially across space or time. For example, some

*Thermoproteales* mediate DSR in thermoalkaline springs, while the same function is likely mediated by *Thermoplasmata* in thermoacidic springs (Fig. 6b,d). This taxonomic variability may result from the unique ecophysiology of these ecologically equivalent organisms that leads them to be selected by different physicochemical characteristics[76,79]. In general, functional redundancy is a common aspect of microbial communities, which is linked to the capacity to withstand environmental pressures and fluctuations.

## Methods

### Sample collection, DNA extraction, sequencing, and metagenomic assembly

This study was conducted in compliance with all relevant ethical standards and regulations. We obtained permits for all sample collection, supported by Yunnan Tengchong Volcano and Spa Tourist Attraction Development Corporation. Geothermal spring sediment samples were collected and transferred with informed consent, under protocols approved by the Ethical Review Committee at the University of Science and Technology of China. A total of 152 sediment samples were collected from 48 geothermal springs spanning six years (2016–2021) located at Tengchong county in Yunnan province, China. These geothermal springs span a wide range of physiochemical parameters with temperature ranging from 23 to 100 °C and pH ranging from 2.0 to 9.7 (Supplementary Data 1). Detailed method for sample collection, DNA extraction, and metagenomic sequencing have been described in Hua et al.[20]. Raw reads generated on Illumina Hiseq 4000 sequencer were preprocessed using custom Perl scripts[80] (https://github.com/hzhengsh/qualityControl). Quality reads for each sample were de novo assembled separately using SPAdes[81] (v.3.9.0) with parameters: -meta -k 21,33,55,77,99,127.

### Genome binning, curation and dereplication

For each sample, quality reads were mapped on assembled scaffolds with length >2500 bp using BBMap (v.38.92; http://sourceforge.net/projects/bbmap/) with following parameters: k = 15 minid = 0.97 build = 1. The generated.bam file was used to calculate sequence depth and further to conduct genome binning using three tools including: MaxBin2[82] (v.2.2.7), CONCOCT[83] (v.1.1.0) and MetaBAT[84] (v.2.12.1). The best bins were determined using DASTool[85] (v.1.1.3). Clean reads for each selected bin were recruited using BBMap (the same parameters as mentioned above) and were reassembled using SPAdes (v.3.15.2) with the following parameters: -careful -k 21,33,55,77,99,127. The genome quality including completeness, contamination, heterogeneity of all bins except small Archaea with tiny sizes and episymbiotic lifestyle (previously known as DPANN Archaea) was evaluated using CheckM[86] (v.1.1.3). For small Archaea bins, their completeness was estimated to calculate the occurrence frequency of genes among a set of 48 single-copy genes[2]. Contamination and heterogeneity of these bins were determined using CheckM. The genome quality for all bins was recorded in Supplementary Data 3. To obtain genomes with high quality, scaffolds with abnormal sequence depth and multiple marker genes were manually removed as previously described[24]. Scaffolds with depth <[quartile$_{1st}$ − 1.5 × (quartile$_{3rd}$ - quartile$_{1st}$)] and > [quartile$_{3rd}$ + 1.5 × (quartile$_{3rd}$ - quartile$_{1st}$)] were treated as outliers. Finally, a total of 2,949 Archaea bins were obtained with completeness > 50% and contamination <10%. The program dRep[87] (v.3.2.2) was applied to dereplicate genomes at 99% ANI (strain level). A total of 603 representative MAGs (rMAGs) were picked out for further metabolic analysis.

### Taxonomic assignment and phylogenetic inference of assembled rMAGs

The taxonomy of all 2949 archaeal genomes was initially determined using GTDB-Tk[88] (v.2.1.0). Phylogenetic analysis was constructed and used to confirm the taxonomic assignments. A total of 603

representative MAGs from present study and reference genomes from GTDB were picked out to infer the phylogeny. Briefly, multiple sequence alignments of 53 concatenated conserved archaeal marker genes were retrieved from GTDB-Tk. The poorly aligned regions were eliminated using trimAl[89] (v.1.4.rev22) with the parameter: -automated1. Phylogeny was inferred using IQ-TREE[90] (v.1.6.12) with 1,000 ultrafast bootstrapping iterations. The best model of LG + F + R10 was determined using ModelFinder[91], which is well supported by Bayesian information criterion (BIC). The phylogenetic tree was visualized using iTOL[92] (v.6).

### Taxonomic assignment of *rpS3* genes

Gene calling was conducted using Prodigal[93] (v.2.6.3) (parameters: -p meta) for scaffolds >500 bp in each metagenome. The candidate *rpS3* protein sequences were retrieved by searching all proteins against curated *rpS3* database (see *rpS3* database at https://doi.org/10.6084/m9.figshare.25650441) which sequences ≥ 200 amino acids using AMPHORA2[94] (v.2.0). To identify *rpS3* protein sequences with accuracy, three steps were performed: (i) Hits with score <40 from AMPHORA-based searches were filtered; (ii) Protein sequences with length <60 amino acids were eliminated; and (iii) All remaining sequences were searched against NCBI-nr database using BLASTp program and only hits identified as *rpS3* genes were kept. Across all metagenomes, a total of 27,436 *rpS3* protein sequences were identified. The confirmed *rpS3* protein sequences were subsequently clustered at 99% sequence identity using USEARCH[95] (v.11.0.667) with following options: -sort length -id 0.99 -maxrejects 0 -maxaccepts 0 -centroids. This resulted in the generation of 8,757 representative archaeal and bacterial *rpS3* proteins at species level.

Since GTDB provided a phylogenetically consistent and rank normalized genome-based taxonomy for prokaryotes, *rpS3* gene sequences from GTDB with well-documented taxonomy information were extracted to build a custom *rpS3* gene database, which could be further used to assign taxonomy at different levels for *rpS3* genes retrieved from metagenomes. Briefly, gene sequences with length >300 bp were kept and were further clustered using USEARCH at 100% identity. To eliminate *rpS3* genes that misassigned to the corresponding genomes due to the biases caused by metagenome assembly and genome binning, taxonomic assignments were conducted for all *rpS3* genes identified from GTDB with themselves as database using assign_taxonomy.py (v.1.9.1) program in QIIME[96]: -m rdp -c 0.60. Items were discarded if the assigned taxonomy is inconsistent with the taxonomy retrieved from GTDB. The retained *rpS3* gene sequences were further clustered at 99% identity using USEARCH to establish a final version of *rpS3* database at species-level. 8,757 representative *rpS3* gene sequences were searched against the curated database with the same procedure as mentioned above. Among them, 2500 *rpS3* genes (28.5%) can't be assigned at phylum-level. These gene sequences were further searched against the protein sequences of curated *rpS3* database using BLASTx program: -evalue 1e-5 -num_descriptions 5 -num_alignments 5. The lowest concordant level of the five top hits was used to assign taxonomy for each undetermined sequence.

### Relative abundance of rMAGs and *rpS3* genes

Clean reads were mapped on 603 rMAGs and scaffolds with length >500 bp using BBMap respectively. The relative abundance of each rMAG in one sample was calculated as the percentage of reads mapped to the rMAG to the total reads mapped to the metagenome. Only rMAG with a relative abundance more than 0.01% is counted. Similarly, clean reads were mapped on 8757 representative *rpS3* gene sequences for each sample using BBMap and the coverage information was calculated by dividing recruited reads to the length of gene length. The relative abundance of each *rpS3* in one sample was calculated by dividing the coverage of one *rpS3* to the total coverage of all *rpS3*. A table was generated to record the abundance information of each

archaeal phylum, class and order based on rMAGs and representative *rpS3* genes in each sample (Supplementary Data 6).

## Genome and community-level metabolic annotation

Functional annotation was conducted using METABOLIC[97] (v.4.0) with all representative archaeal MAGs and clean reads of each sample as inputs. Briefly, gene calling was performed for each rMAG using Prodigal with "-p single" option to identify open reading frames (ORFs). ORFs were subsequently searched against the KEGG, TIGRfam, Pfam and collected HMM profiles associated with key biogeochemical cycling using HMMER[98] (v.3.3.2). The MW-scores (metabolic weight score) were calculated by METABOLIC which describes the contribution of one function/pathway relative to all functions in all Archaea since only archaeal genomes were taken as inputs.

The metabolic pathway associated with elemental biogeochemical cycling were further assessed by searching all ORFs in rMAGs against the KOfam database (KEGG release 103) using the KofamScan[99] (v.1.3.0; E value < $10^{-5}$). The archaeal order was determined to contain the pathway if all genes involved in this pathway can be detected in at least one rMAG in a given order. To ensure the accuracy, only rMAGs with scaffold numbers <200 were taken into consideration. The gene repertoire against Carbohydrate-Active enZYmes (CAZy) Database[100] for searching carbohydrate degradation capacities was explored for each rMAG as well.

## Phylogenetic tree reconstruction of hydrogenase

Hydrogenase proteins were initially identified using METABOLIC across 603 archaeal rMAGs in this study, and were further classified into different functional groups using a web tool of Hydrogenase database (HydDB)[60]. Reference sequences of [Fe], [FeFe] and [NiFe] hydrogenases in Archaea were downloaded from the HydDB. The obtained protein sequences were aligned using MUSCLE[101] (v.3.8.31) and trimmed using TrimAL (-gt 0.05 -cons 50). Phylogeny was generated using IQ-tree with 1,000 ultrafast bootstrapping iterations and VT + F + R9 as the best model.

## Metatranscriptomic analysis

Geothermal spring sediment samples collected in Jan 2021 and Jul 2022 were conducted for metatranscriptomic sequencing. Sufficient RNA was only obtained from four samples, including JZ-2 (pH 7.47, 36.0 °C) in Jan 2021 and DRTY-16 (pH 2.11, 38.1 °C), QQ (pH 6.80, 70.1 °C), JZ-3 (pH 9.98, 64.7 °C) in Jul 2022. Total RNA was extracted using the RNeasy PowerSoil Total RNA kit (QIAGEN). Total RNA was subsequently sent out for metatranscriptomic sequencing, following with rRNA removal and mRNA enrichment. The generated paired-end reads ( ~114 Gbp) were quality-controlled using Sickle (v.1.33; https://github.com/najoshi/sickle) with the following parameters: -t sanger --quiet -l 50. Trimmed transcripts were mapped to the identified ORFs of 603 archaeal rMAGs using BBMap with the same parameters as above. The fragments per kilobase of transcript per million (FPKM) values were extracted from the output. The heatmap figure was built based on gene expression data using the pheatmap package (v.1.0.12) in RStudio (v.1.2.5033).

## Statistical analyses

A total of 132 geothermal spring samples with both pH and temperature measured were picked out and applied for ordination analysis. To assess the resemblance of samples and to identify main factors affecting the clustering pattern, principal coordinates analysis (PCoA) was conducted based on Bray-Curtis distance of archaeal community composition and functional annotation (KOs) data. Three factors were considered including sampling season (summer and winter), pH (acidic: pH <5.5; neutral: 5.5 ≤ pH <8.5; alkaline: pH ≥8.5)[33], and temperature (mesothermal: temperature <60 °C; thermal: 60 ≤ temperature <80 °C; hyperthermal: temperature ≥80 °C)[34]. Analysis of

Similarity (ANOSIM) was used to examine the significant differences among different sample groups. Pearson correlation was applied to test the correlation between pH and temperature with the relative abundance of specific archaeal lineages. The generalized linear model (GLM) was applied to establish connections between archaeal community at strain/function level and environmental parameters using the *glm* function in the "stats" package (v. 4.2.2)[102] in R platform (http://cran.r-project.org). Statistical significance ($p < 0.05$) was assessed by conducting likelihood ratio tests (LRT) on the linear models, utilizing Type-II ANOVA in the "car" package (v. 3.0-10)[103] in R.

## Network-based co-occurrence analysis

A comprehensive dataset comprising 48 acidic and 31 alkaline geothermal springs, spanning both temporal and spatial dimensions, was employed for unveiling the co-occurrence patterns of archaeal orders (Supplementary Data 1). To enhance clarity and streamline the analysis, archaeal rMAGs occurring in fewer than half of samples were excluded, resulting in the retention of 83 rMAGs for subsequent network construction. The co-occurrence networks were constructed using SparCC[104] with default parameters, and the robustness of the networks was ensured by employing 100 bootstrap samples to infer pseudo-P values. Only correlations deemed statistically significant ($p < 0.05$, two-sided) and robust (absolute value of correlation coefficient greater than 0.6) between pairwise rMAGs were considered, contributing to the inference of a reliable network. Visualization of the network was performed using Gephi (v.0.10.1)[105].

## Reporting summary

Further information on research design is available in the Nature Portfolio Reporting Summary linked to this article.

## Data availability

2949 metagenome-assembled genomes of Archaea described in this study have been deposited in NCBI under the BioProject PRJNA544494: BioSample id SAMN18253264 to SAMN18253267, SAMN18253269, SAMN18253270, SAMN18838809, SAMN19656016 to SAMN19656018, SAMN28867992 to SAMN28867995, SAMN28867997 to SAMN28867999, SAMN31028420 to SAMN31028426, SAMN31028428 to SAMN31028439, SAMN31028763, SAMN34195732, SAMN36035244 to SAMN36035357. The accession numbers of genomes are JAVYKE000000000 to JAWCTO000000000, with detailed accession for each MAG recorded in Supplementary Data 3. Supplementary data, comprising the phylogenies of Archaea and hydrogenase, the *rpS3* protein sequences, and co-occurrence networks between Archaea and Bacteria, are accessible in the FigShare repository at https://doi.org/10.6084/m9.figshare.25650441. The links to the databases used in this study are listed below: CAZy [https://www.cazy.org/]; Genome Taxonomy database [https://data.ace.uq.edu.au/public/gtdb/data/releases/release207/]; HMM database within METABOLIC program (v.4.0) [https://github.com/AnantharamanLab/METABOLIC/releases].

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

## Acknowledgements

We thank Guangdong Magigene Biotechnology Co., Ltd. China for the assistance in data analysis, and the entire staffs from Yunnan Tengchong Volcano and Spa Tourist Attraction Development Corporation for strong support. This work was financially supported by National Natural Science Foundation of China (42207145, Y.L.Q.; 32170014, Z.S.H.; 42377312, Y.T.C.) and the U.S. National Science Foundation (DBI 1557042, B.P.H.).

## Author contributions

Y.L.Q., Y.T.C., W.S.S., W.J.L., B.P.H. and Z.S.H. conceived the study. Y.L.Q., Y.X.L., Y.Z.R., M.M.L., L.C., J.Y.J. and Z.S.H. performed the sample collection. Y.L.Q., Y.X.L., Y.Z.R., Z.C.X., Z.X.Y., and M.M.L. performed the measurement of physiochemical parameters, DNA and RNA extraction. Y.L.Q., Y.T.C., Y.G.X., Y.X.L., Y.Z.R., Q.J.X., X.R.C., Y.N.Q., L.L. and Z.S.H. performed the metagenomic and metatranscriptomic analyses, genome binning, functional annotation, and evolutionary analysis. Y.L.Q., Y.T.C., W.J.L., W.S.S., B.P.H and Z.S.H. wrote the manuscript. All authors discussed the results and commented on the manuscript.

## Competing interests

The authors declare no competing interests.
