## [Peer Review File · Nature Communications]

REVIEWER COMMENTS

Reviewer #1 (Remarks to the Author):

The authors analyzed numbers of archaeal MAGs (metagenome-assembled genomes) from diverse hot springs in China using in combination of metagenomics and metatranscriptomics. The metabolic potentials of the diverse archaeal lineages are quite interesting. However, I have found issues listed below.

1. The absence of essential geochemical parameters that are required in geochemical cycles such as hydrogen, methane, sulfide, ammonia (ammonium), nitrate etc.
2. The lack of dataset of bacterial population that is also essential to understand geochemical cycles.

Minor comments

L87-: Microbial cell abundance or alternative quantitative data is also essential to evaluate microbial communities.

L108- , L138 and through the manuscript: Which taxonomy did the authors use, e.g. GTDB through the manuscript, or some were GTDB and some were not? Please clearly mention about the taxonomy used in this study.

L108-110: Bacterial composition in combination with cell abundance is also essential to evaluate the quality of sampling and library construction because some of the bacterial lineages are signatures of contamination during the processes.

L114: TACK is confusing especially considering the broad readership of this journal. The name of the three phyla have already disappeared in the current taxonomy. If the authors prefer to use, appropriate explanation and/or references must be required.

L115: Please take care about DAPANN as well.

L138-: Do the authors have a plan to give appropriate taxonomic names for lineages presented in alphabet and numbers via SeqCode?

L147: Euryarchaeota is no longer validated, and probably Methanobacteriota (Göker and Oren, *Int. J. Syst. Evol. Microbiol.* 2023;73:006024. DOI 10.1099/ijsem.0.006024)

L214: “genes encoding fermentations” is not appropriate in most of the cases. “genes related to fermentation pathways” or “genes related to substrate-level phosphorylation” would be better.

L217-218: If a gene for POR is encoded, the organism presumably harbors a system for energy conservation in addition to substrate-level phosphorylation. Thus, I do not prefer to use fermentation that have diverse meaning.

L224-226: The structure of the paragraph is quite confusing for readers. Just above the sentence, the issues related to substrate-level phosphorylation including acetate production are discussed.

L226 (Additionally-): The discussion is too speculative. The reaction is reversible. Thus, such issue should be discussed after identification of the metabolic pathway in reliable MAGs but not using the specific gene.

L237-239: The sentence is very confusing. The presence of Sulfolobales in alkaline hot spring and harbor DC/4HB pathway are quite surprising.

L368- (especially, 382-385): The discussion about species interaction among the archaeal population is too much speculative because metabolic potential of bacterial population is not presented in this manuscript.

L393-: The metabolic function of low abundance population may also be overestimated. The authors should also pay attention about the presence of bacterial population with similar metabolic potential.

Reviewer #2 (Remarks to the Author):

This manuscript by Qi et al. analyses ca. 3k archaeal genomes from terrestrial geothermal springs in China. Many springs over a significant period of time (ca. 6 years) have been sampled and their DNA and RNA have been analysed using state-of-the art approaches. The authors found that temperature and pH are likely the key drivers for shaping the community structure and community-wide gene expression to cope with the distinct environmental conditions in these extreme environments.

To the best of my knowledge, this is the most extensive study ever done in hot spring ecosystems and with a focus on archaea. Overall, the main results are interesting and well presented. Despite my excitement about this study, I have a few suggestions to make to improve the quality of this paper.

1) Although the authors have sequenced RNA, the manuscript barely refers to metatranscriptomes and I wonder why that is. I think those results are important for underpinning the metabolic discussions especially in the context of adaptation to different springs with distinct env. conditions. Hence, I suggest to map the transcripts to the MAGs and also showing them independently from the MAGs as some of them might not be mapping.

2) This manuscript would benefit from co-occurrence analyses (e.g. which MAGs do co-occur?) in addition to gene network analyses (e.g. WGCNA). Currently, the paper is too much focussing on known pathways and their regulation. Thus, having a broader context will put those results into perspective and it will support some of the hypotheses (e.g., metabolic interaction between MAGs). For instance, those results are needed to provide evidence for the speculations between lines 377 and 407. Without evidence, this paragraph is too speculative and should therefore be deleted.

3) I suggest to run a generalised linear model (GLM) with all environmental variables to build a relationship between the response (e.g. metabolic genes, MAGs) and predictors (e.g. env. variables) (e.g. lines 156-172). This will provide a more comprehensive approach for assessing which environmental variables are most important for the distinct metabolism identified in different springs. As many env. variables as possible should be included in this analyses.

REVIEWER COMMENTS

Reviewer #1 (Remarks to the Author):

The authors analyzed numbers of archaeal MAGs (metagenome-assembled genomes) from diverse hot springs in China using in combination of metagenomics and metatranscriptomics. The metabolic potentials of the diverse archaeal lineages are quite interesting. However, I have found issues listed below.

Response: We thank the reviewer for the kind assessment of our work.

Comment 1. The absence of essential geochemical parameters that are required in geochemical cycles such as hydrogen, methane, sulfide, ammonia (ammonium), nitrate etc.

Response: Thank you for your valuable suggestions. We have incorporated relevant geochemical parameters for Tengchong geothermal springs in Supplementary Data 1. These include EC, salinity, Cl^- , organic matter, total organic carbon, sulfate, ammonia, nitrate, nitrite, total nitrogen and total phosphorus. Regrettably, the concentrations of hydrogen and methane were not measured in our study.

Comment 2. The lack of dataset of bacterial population that is also essential to understand geochemical cycles.

Response: We acknowledge the importance of bacterial populations in understanding geochemical cycles in geothermal ecosystems. Our study, however, intentionally focuses on archaeal communities, given their limited exploration compared to Bacteria. This has gained positive assessment by the second reviewer who wrote “To the best of my knowledge, this is the most extensive study ever done in hot spring ecosystems with a focus on Archaea. Overall, the main results are interesting and well presented.” Additionally, we observed a higher relative abundance of Archaea than Bacteria in 54 of the geothermal spring samples, particularly under polyextreme conditions (e.g., thermoacidic and thermoalkaline springs), underscoring the significance of Archaea in these systems.

The dominance of Archaea over Bacteria in such extreme environments offers an important opportunity to explore the ecological roles of these less studied microbial groups. To mitigate concerns about overstating the contributions of Archaea, we shifted our focus to those communities predominated by Archaea (Fig. 6; Lines 437 – 490). Also, we have incorporated co-occurrence network analyses to more objectively analyze potential functional interactions among Archaea. By concentrating on Archaea, our study aims to provide a deeper insight into the potential ecological roles, and functional interactions among extremophilic Archaea. We believe our work makes a substantial contribution to the field of extremophile research.

Minor comments

Comment 3. L87-: Microbial cell abundance or alternative quantitative data is also essential to evaluate microbial communities.

Response: Regrettably, we lack specific microbial cell abundance data or alternative quantitative information for our previous 152 metagenomic samples. Instead, we performed qPCR experiments on our recent samples collected in Aug, 2023 to confirm the high abundance of Archaea (see Table 1 below). Universal primers (updated primers 515F (Parada) and 806R (Apprill)¹ from the Earth Microbiome Project) were applied to quantify the total cells of both Bacteria and Archaea, while Archaea-specific primers (Arch519F and Arch915R) were used to determine the cell counts of Archaea. Overall, many samples exhibited a high abundance of Archaea, consistent with our metagenomic findings. For instance, the relative abundance of Archaea in samples DRTY-2, DRTY-3 and DRTY-19 as determined by qPCR (96.4%, 78.7%, 44.3%, respectively) mirrors the estimates from metagenomic datasets (~97.7%, 72.5%, 45.7%, respectively; Figure 1a in Manuscript). While there are some variations between results obtained from the two techniques. For example, the relative abundance of Archaea in JZ-1, JZ-2, JZ-4 and QQ as determined by qPCR, is 77.9%, 13.1%, 12.3% and 26.4% respectively, slightly higher than the highest abundance of 54.5%, 6.9%, 7.4% and 20.1% in metagenomic datasets. The qPCR result for the DRTY-9 sample (57.4%) indicates a lower abundance of Archaea compared to the lowest abundance based on metagenomic data (74.7%). We consider these variations acceptable, as they are samples collected at different times. Variations within the same sampling site over different times are normal. We also conducted amplicon sequencing of the 16S rRNA gene (V4 region; 515F (Parada) and 806R (Apprill)) on recently collected samples in February 2023. The obtained data revealed that Archaea constitute an average abundance of 36.6% and a maximum abundance of 90.4% (see Figure 1 below). Additionally, acidic DRTY springs and alkaline ZMQ springs were dominated by Archaea. The observed prevalence and abundance of Archaea align with the conclusions drawn from our study.

Previous published studies also report a high abundance of Archaea in some of the specific springs we studied. Li *et al.* (2015) applied CARD-FISH to explore the community composition of the SRBZ pool². They reported that Archaea were more abundant in samples with higher temperatures (74.6 to 90.8°C), reaching peak abundance at 90.8°C and constituting more than 90% of the total cell count. Additionally, Hou *et al.* (2013) conducted a survey on microbial diversity of 37 Tengchong geothermal springs using 16S rRNA gene pyrosequencing³. They reported the similar community composition to ours, where the archaeal order *Sulfolobales* dominated in high-temperature and acidic springs (ZZQ and DRTY-1 pools), comprising over 82% of all sequences. In high-temperature springs with circumneutral to alkaline pH (GMQ, JZ, SRBZ, and ZMQ pools), *Sulfolobales* and *Thermoproteales* were the dominant Archaea, accounting for approximately 5% to 95% of sequences in those microbial communities. *Nitrososphaerales* predominated among the Archaea in GXS spring, characterized by moderate temperature and neutral pH, representing approximately 10% of total sequences. The order *Thermoplasmatales* emerged as the predominant Archaea in the lowest-temperature acidic DRTY-3 spring, constituting over 10% of the total sequences.

Collectively, the aforementioned information regarding the abundance of Archaea aligns with the results of our study, highlighting the diverse and abundant Archaea resources present in Tengchong geothermal springs.

Sample	Microbial 16S rRNA gene (EMP-F, EMP-R)		Archaeal 16S rRNA gene (Arch519F, Arch915R)		Relative abundance of Archaea	
	Ct	Absolute copy number/ μ l	Ct	Absolute copy number/ μ l	qPCR	Metagenome
DRTY-2	23.9	1.10E+06	19.8	1.06E+06	96.4%	57.1 – 97.7% (ave. 83.1%)
DRTY-3	29.7	2.67E+04	25.3	2.10E+04	78.7%	56.4 – 87.1% (ave. 72.5%)
DRTY-9	25.5	3.94E+05	22.0	2.26E+05	57.4%	74.7 – 97.8% (ave. 86.9%)
DRTY-19	24.7	6.84E+05	21.6	3.03E+05	44.3%	34.5 – 56.9% (ave. 45.7%)
JZ-1	26.2	1.10E+07	26.2	8.55E+06	77.9%	10.1 – 58.3% (ave. 36.7%)
JZ-3	19.6	1.80E+07	18.7	2.35E+06	13.1%	0.1 – 6.9% (ave. 2.9%)
JZ-4	21.6	5.01E+06	20.6	6.14E+05	12.3%	0.2 – 7.4% (ave. 2.7%)
QQ	21.9	4.17E+06	19.7	1.10E+06	26.4%	3.1 – 20.1% (ave. 9.6%)

Table 1. Relative abundance of Archaea determined by qPCR experiments. Samples were collected from Tengchong geothermal springs in Aug, 2023.

Figure 1. Relative abundances of Archaea and Bacteria in microbial community determined through 16S rRNA-based amplicon sequencing. Samples were collected from Tengchong geothermal springs in February, 2023.

Comment 4. L108- , L138 and through the manuscript: Which taxonomy did the authors use, e.g. GTDB through the manuscript, or some were GTDB and some were not? Please clearly mention about the taxonomy used in this study.

Response: We apologize for any confusion that may have arisen from the use of informal names such as TACK and ASGARD. For simplicity, we consistently applied the GTDB taxonomy throughout the entire revised manuscript. In some cases, we also mention other taxonomic names along with GTDB names in order to better connect with the literature.

Comment 5. L108-110: Bacterial composition in combination with cell abundance is also essential to evaluate the quality of sampling and library construction because some of the bacterial lineages are signatures of contamination during the processes.

Response: We need to emphasize that the representative *rpS3* sequences recovered from metagenomic data encompass sequences from both Archaea and Bacteria (refer to **Methods**, Lines 640 – 644). The archaeal composition and relative abundance calculated based on the representative *rpS3* sequences are both highly similar to those calculated using rMAGs (both bacterial and archaeal MAGs were reconstructed but only archaeal MAGs were described), as illustrated in Figure 1b and Supplementary Figure 1. Detailed abundance information for each archaeal phylum, class, and order, based on both rMAGs and representative *rpS3* genes in each sample, is provided in Supplementary Data 6. Also, as mentioned in Comment 3, we applied different strategies to evaluate the relative abundance of Archaea. Results consistently supported the high abundances of Archaea in many of the geothermal spring samples. This comprehensive approach reinforces the reliability of our study and ensures a thorough assessment of sampling and library construction quality.

Comment 6. L114: TACK is confusing especially considering the broad readership of this journal. The name of the three phyla have already disappeared in the current taxonomy. If the authors prefer to use, appropriate explanation and/or references must be required.

Response: TACK has been replaced with *Thermoproteota* throughout our entire manuscript. As mentioned above, the revised manuscript uses the current GTDB taxonomy.

Comment 7. L115: Please take care about DPANN as well.

Response: Done as suggested. We now use terms like "small Archaea" "Archaea with small genome sizes", "Archaea with episymbiotic lifestyle" or detailed taxonomic information to replace the usage of DPANN, providing more specific and descriptive language.

Comment 8. L138-: Do the authors have a plan to give appropriate taxonomic names for lineages presented in alphabet and numbers via SeqCode?

Response: We are aware that we can name many new Archaea under the SeqCode. We discussed this opportunity carefully as an author team and we chose not to do this. We believe naming should be reserved for papers that are focused on the properties of lower ranks because such studies would be better suited to provide more detailed information about the organisms belonging to those lower ranks. Our paper covers all Archaea in these geothermal springs and as such we don't focus enough on any particular group of Archaea to justify naming, in our opinion. We do hope that our work can be used in future, more taxonomically focused papers to support naming some of these Archaea.

Comment 9. L147: Euryarchaeota is no longer validated, and probably Methanobacteriota (Göker and Oren, *Int. J. Syst. Evol. Microbiol.* 2023;73:006024. DOI 10.1099/ijsem.0.006024)

Response: We have discarded the name "*Euryarchaeota*" and adopted a classification based on the

GTDB, including *Hadarchaeota*, *Methanobacteriota*, *Halobacteriota*, and *Thermoplasmatota* in this study.

Comment 10. L214: “genes encoding fermentations” is not appropriate in most of the cases. “genes related to fermentation pathways” or “genes related to substrate-level phosphorylation” would be better.

Response: Revised as suggested, including here and elsewhere (Lines 256, 581, 1001).

Comment 11. L217-218: If a gene for POR is encoded, the organism presumably harbors a system for energy conservation in addition to substrate-level phosphorylation. Thus, I do not prefer to use fermentation that have diverse meaning.

Response: We fully concur with your opinion and relevant description has been revised accordingly.

Comment 12. L224-226: The structure of the paragraph is quite confusing for readers. Just above the sentence, the issues related to substrate-level phosphorylation including acetate production are discussed.

Response: To improve clarity, we restructured the paragraph to first describe acetate oxidation and then other metabolisms, including fermentation and pyruvate metabolism (Lines 245 – 274).

Comment 13. L226 (Additionally-): The discussion is too speculative. The reaction is reversible. Thus, such issue should be discussed after identification of the metabolic pathway in reliable MAGs but not using the specific gene.

Response: We agree with the reviewer that this sentence is a bit speculative. Due to the wide detection of ACD in many Bacteria, we have removed the citation in the main text. Thus, we have rephrased the sentence to: *Additionally, the widespread existence and expression of acetate-CoA ligases (acdAB; Fig. 4 and Supplementary Data 9) indicates that many Archaea can oxidize acetyl-CoA for energy conservation or, in reverse, consume acetate.* (Lines 266 – 268)

Comment 14. L237-239: The sentence is very confusing. The presence of Sulfolobales in alkaline hot spring and harbor DC/4HB pathway are quite surprising.

Response: We understand the reviewer has two main concerns: 1) How can *Sulfolobales* can be abundant in alkaline geothermal springs? 2) Whether the DC/4HB carbon fixation pathway could exist in *Sulfolobales*.

Regarding the first concern, it's crucial to note that *Sulfolobales* is an order comprising diverse taxa (seven families in our study). Many well-known members of the order favor acidic environments, such as *Sulfolobus*, *Acidianus*, and *Acidilobus*. Nevertheless, our study reveals a high abundance of members of the order *Sulfolobales* in alkaline geothermal springs (see Table 2 below). Generally, *Sulfolobaceae* and *Fervidicoccaceae* are more abundant in acidic hot springs, while *Desulfurococcaceae* prefers alkaline springs. Also, note that *Desulfurococcaceae* is a family in the

order *Desulfurococcales* according to the LPSN and NCBI taxonomy, but is a family in the order *Sulfolobales* according to the GTDB taxonomy. Our results also show that some members of the *Sulfolobaceae* (e.g., DRTY-9_201705_bins_5) are abundant in both acidic and alkaline springs. A 16S rRNA gene pyrosequencing analysis published in 2013 reported the high abundance of *Desulfurococcales* in Tengchong springs with circumneutral to alkaline pH (pH: 6.7 – 9.4)³. This research aligns with our findings, affirming the high abundance of some *Sulfolobales* to circumneutral and alkaline hot springs.

Regarding the second concern, we apologize for our inaccurate statement. Actually, some members of the order *Sulfolobales* have the capacity to fix carbon dioxide through either the 3HP/4HB pathway or DC/4-HB pathway. Specifically, two rMAGs (DRTY-2_202007_bins_22, DRTY-9_201705_bins_5) harbor a complete set of genes for the 3HP/4HB pathway, and one rMAG (DRTY-4_201912_bins_2) harbors the complete set of genes for the DC/4-HB pathway (Supplementary Data 8). Notably, DRTY-2_202007_bins_22 and DRTY-4_201912_bins_2 are prevalent in acidic springs, while DRTY-9_201705_bins_5 is abundant in both acidic and alkaline springs. In contrast, several rMAGs within the *Thermoproteales* that are abundant over wide pH ranges encode the DC/4-HB pathway. Furthermore, the enzymatic activities associated with both the 3HP/4HB and DC/4-HB pathways in representative *Sulfolobales* have been experimentally confirmed and reported previously⁴. To avoid confusion, we have rephrased the sentence to: *Thermoproteales* encoding a complete set of genes involved in the dicarboxylate-hydroxybutyrate (DC/4-HB) cycle including the key enzymes acetyl-CoA/propionyl-CoA carboxylase (K01964, K15036, K15037) were abundant in both acidic and alkaline springs (Supplementary Data 8). Additionally, in acidic springs, members of *Sulfolobales* were abundant and encoded either the 3-hydroxypropionate/4-hydroxybutyrate (3HP/4HB; with pyruvate synthase and PEP carboxylase being the key enzymes (K01007, K01595)) or DC/4-HB cycles, as supported by the detection of a complete set of genes for both pathways. Surprisingly, some members of the *Sulfolobales* were highly abundant in alkaline springs but only possessed the 3HP/4HB cycle, suggesting that 3HP/4HB might also be the primary pathway for carbon fixation in such environments. (Lines 278 – 286).

Sample	pH	Acidilobaceae	AG1	Desulfurococcaceae	Feravidicoccaceae	Ignisphaeraceae	NBVN01	Sulfolobaceae
201709_DRTY-7	2	0.00%	0.07%	0.01%	2.49%	0.00%	0.00%	2.50%
201912_DRTY-7	2.19	0.00%	0.07%	0.04%	0.05%	0.01%	0.18%	58.81%
202101_DRTY-10	2.19	0.00%	0.05%	0.00%	0.15%	0.00%	0.03%	11.70%
202007_DRTY-6	2.27	0.00%	0.06%	0.02%	0.68%	0.02%	0.04%	10.67%
201912_DRTY-10	2.3	0.00%	0.30%	0.04%	0.43%	0.01%	0.11%	77.24%
202101_DRTY-9	2.38	0.00%	0.04%	0.00%	0.22%	0.00%	0.01%	4.63%
201912_DRTY-4	2.4	0.00%	0.59%	0.04%	0.26%	0.01%	0.29%	86.24%
201912_DRTY-5	2.45	0.00%	0.23%	0.04%	8.87%	0.01%	0.55%	66.73%
201709_DRTY-9	2.5	0.00%	0.14%	0.01%	2.36%	0.00%	0.00%	1.71%
202007_DRTY-3	2.55	0.00%	0.01%	0.00%	0.73%	0.00%	0.00%	2.22%
201912_DRTY-6	2.58	0.00%	0.03%	0.06%	1.19%	0.02%	0.05%	4.59%
201912_DRTY-3	2.76	0.00%	0.02%	0.00%	0.71%	0.00%	0.01%	3.02%
202007_DRTY-2	2.86	0.00%	0.18%	1.05%	7.74%	1.16%	0.39%	9.64%
201803_DRTY-7	3	0.00%	0.04%	0.13%	0.69%	0.02%	0.06%	13.80%
201808_DRTY-9	3	0.00%	0.04%	0.01%	0.78%	0.00%	0.00%	1.44%
201901_DRTY-4	3	0.00%	0.46%	0.01%	0.63%	0.00%	0.17%	75.12%
201907_DRTY-6	3	0.00%	0.13%	0.09%	1.82%	0.02%	0.02%	3.96%
201907_DRTY-7	3	0.00%	0.02%	0.05%	0.03%	0.00%	0.01%	75.42%
201912_DRTY-2	3	0.00%	0.07%	3.88%	31.96%	2.83%	1.39%	13.20%
201901_DRTY-9	3.04	0.00%	0.09%	0.01%	0.29%	0.00%	0.03%	9.52%
201912_DRTY-1	3.4	0.00%	0.14%	0.10%	22.16%	0.01%	0.11%	18.81%
201901_DRTY-3	3.52	0.00%	0.05%	0.02%	1.27%	0.00%	0.01%	3.34%
201901_DRTY-1	3.53	0.02%	5.42%	1.55%	16.36%	5.44%	0.77%	13.59%
202101_ZZQ	3.6	0.00%	13.01%	0.07%	0.01%	0.05%	0.09%	62.58%
201901_DRTY-2	3.62	0.01%	1.43%	0.79%	6.27%	0.82%	0.95%	14.05%
201705_DRTY-3	4	0.00%	0.02%	0.01%	0.49%	0.01%	0.00%	1.92%
201709_DRTY-3	4	0.00%	0.02%	0.02%	1.30%	0.00%	0.00%	1.21%
201803_DRTY-5	4	0.00%	0.16%	0.02%	7.28%	0.00%	0.01%	19.31%
201803_DRTY-9	4	0.00%	0.05%	0.00%	0.41%	0.00%	0.00%	1.65%
201808_DRTY-1	4	0.01%	6.00%	1.33%	8.51%	6.23%	0.38%	48.45%
201808_DRTY-2	4	0.00%	0.04%	0.00%	9.59%	0.00%	0.00%	1.67%
201808_DRTY-3	4	0.00%	0.03%	0.02%	1.70%	0.00%	0.01%	1.53%
201808_DRTY-5	4	0.00%	0.23%	0.02%	8.79%	0.00%	0.18%	29.11%
201907_DRTY-3	4	0.00%	0.02%	0.05%	1.70%	0.00%	0.01%	3.47%
201705_DRTY-1	4.5	0.00%	0.10%	0.17%	27.62%	0.13%	0.09%	4.03%
201803_DRTY-3	5	0.00%	0.02%	0.01%	1.95%	0.00%	0.00%	0.34%
201803_DRTY-8	5	0.02%	4.24%	33.43%	1.37%	0.98%	0.18%	1.59%
201808_DRTY-8	5	0.13%	24.43%	12.76%	1.28%	1.78%	1.63%	12.33%
201907_DRTY-5	5	0.00%	0.16%	0.17%	7.44%	0.00%	0.11%	20.37%
201601_ZZQ	5.4	0.00%	25.31%	0.19%	0.09%	0.05%	0.11%	56.67%
201803_ZZQ	6	0.02%	0.06%	1.13%	0.00%	1.75%	2.17%	0.00%
201601_DRTY-7	6.02	0.01%	1.57%	2.37%	3.19%	0.13%	0.19%	0.06%
201601_JZ-2	6.5	1.56%	0.35%	1.80%	0.03%	1.92%	1.05%	0.04%
201901_JZ-1	6.57	0.08%	0.16%	0.63%	0.01%	0.64%	1.50%	0.03%
201912_JZ-1	7.13	0.03%	0.07%	0.79%	0.04%	0.07%	1.60%	0.28%
201901_SRBZ-1	7.2	0.10%	0.16%	5.83%	0.41%	0.93%	2.67%	0.03%
201705_JZ-1	7.3	0.15%	0.10%	0.86%	0.00%	2.22%	2.30%	0.00%
201901_ZZQ	7.35	0.02%	7.02%	4.02%	0.19%	1.42%	0.80%	0.17%
201912_GXS	8.36	0.00%	0.12%	0.78%	0.03%	0.12%	1.76%	0.23%
201709_JZ-1	8.5	0.39%	0.13%	1.43%	0.02%	1.01%	2.09%	0.00%
201709_SZTDM-1b	8.5	1.07%	0.72%	7.22%	0.01%	0.13%	1.07%	0.00%
201709_SZTDM-1c	8.5	0.14%	0.24%	4.14%	0.23%	0.16%	1.71%	0.00%
201901_ZMQR	8.5	9.94%	0.20%	33.99%	0.01%	1.10%	2.54%	0.00%
201705_DRC	8.7	0.26%	0.21%	4.89%	0.07%	0.09%	0.31%	0.02%
202007_SRBZ-1	8.73	1.54%	0.20%	1.84%	0.00%	0.15%	1.19%	0.00%
201601_GMQS	9	0.84%	0.21%	12.60%	0.01%	2.68%	8.25%	0.04%
201709_SRBZ-2	9	0.00%	0.61%	7.13%	0.00%	0.24%	0.10%	0.00%
201803_GMQP	9	0.18%	0.27%	11.98%	0.01%	1.61%	3.39%	0.01%
201803_SRBZ-1	9	0.07%	0.48%	6.11%	0.00%	0.11%	0.29%	0.00%
201803_SRBZ-2	9	0.15%	0.40%	19.26%	0.01%	0.46%	7.90%	0.00%
201803_ZMQR	9	9.15%	0.23%	30.33%	0.01%	1.66%	2.05%	0.00%
201808_GMQP	9	0.11%	0.10%	6.93%	0.02%	1.29%	6.30%	0.04%
201808_SRBZ-1	9	0.02%	0.41%	6.74%	0.71%	1.00%	21.39%	0.00%
201808_ZMQR	9	5.60%	0.26%	34.91%	0.03%	1.39%	2.90%	0.06%
201907_SRBZ-1	9	0.70%	0.39%	10.56%	0.00%	0.51%	0.19%	0.00%
202101_GMQP	9.43	0.30%	0.23%	10.61%	0.01%	1.03%	3.25%	0.00%
201705_ZMQR	9.5	3.62%	0.10%	29.28%	0.02%	6.64%	3.83%	0.00%
201709_GMQP	9.5	0.18%	0.35%	7.90%	0.00%	1.37%	1.70%	0.00%
201709_GMQS	9.5	1.63%	0.02%	27.03%	0.01%	0.60%	1.27%	0.00%
201709_ZMQL	9.5	1.58%	1.03%	11.35%	0.02%	1.68%	1.27%	0.02%
201709_ZMQR	9.5	4.11%	0.09%	22.29%	0.01%	1.40%	1.20%	0.00%
201907_GMQP	9.5	0.38%	0.45%	9.85%	0.00%	0.47%	1.18%	0.00%
201907_GMQS	9.5	1.26%	0.00%	22.95%	0.01%	0.30%	0.56%	0.00%
201912_GMQP	9.58	2.19%	0.14%	27.29%	0.05%	4.45%	7.61%	0.27%
201705_GMQP	9.6	0.16%	0.15%	10.93%	0.01%	4.07%	6.04%	0.00%
201912_ZMQR	9.71	2.55%	0.27%	17.67%	0.04%	0.69%	3.59%	0.30%
201912_ZMQL	9.73	0.01%	12.66%	0.10%	0.05%	0.02%	0.12%	70.76%

Table 2. Relative abundance of families from the order *Sulfolobales* in the microbial community in this study (only samples with a maximum relative abundance exceeding 1% are shown)

Comment 15. L368- (especially, 382-385): The discussion about species interaction among the archaeal population is too much speculative because metabolic potential of bacterial population is not presented in this manuscript.

Response: We agree with the reviewer that the discussion regarding potential functional interactions among Archaea is too speculative without considering Bacteria. A similar comment was also raised by the second reviewer (comment 2). To address this concern, we made some changes in our Revised Manuscript. Firstly, we now specifically focus on investigating potential functional interactions among Archaea only in communities where Archaea predominate (> 50% relative abundance), particularly under polyextreme conditions (e.g., thermoacidic and thermoalkaline springs) (Lines

443 – 445). Secondly, we have introduced a co-occurrence network analysis in the Revised Manuscript, as suggested by Reviewer #2 (Fig. 6, Lines 723 – 732 in **Methods**). This analysis enabled us to examine potential functional interactions among co-occurring Archaea that demonstrate positive or negative correlations. The co-occurrence analysis provides a more empirical and data-driven foundation for a better discussion of potential species interactions within the archaeal communities.

Comment 16. L393-: The metabolic function of low abundance population may also be overestimated. The authors should also pay attention about the presence of bacterial population with similar metabolic potential.

Response: We acknowledge that our previous manuscript probably overestimated potential metabolic interactions between low-abundance Archaea, especially because we did not account for Bacteria. In the Revised Manuscript, we removed the previous paragraph and provide a new section that focuses more on abundant archaea entitled “Some Archaea may be competitive or cooperative” (Lines 437 – 504).

Reviewer #2 (Remarks to the Author):

This manuscript by Qi et al. analyses ca. 3k archaeal genomes from terrestrial geothermal springs in China. Many springs over a significant period of time (ca. 6 years) have been sampled and their DNA and RNA have been analysed using state-of-the art approaches. The authors found that temperature and pH are likely the key drivers for shaping the community structure and community-wide gene expression to cope with the distinct environmental conditions in these extreme environments.

To the best of my knowledge, this is the most extensive study ever done in hot spring ecosystems and with a focus on archaea. Overall, the main results are interesting and well presented. Despite my excitement about this study, I have a few suggestions to make to improve the quality of this paper.

Response: We sincerely thank the reviewer for the positive assessment and valuable comments!

Comment 1. Although the authors have sequenced RNA, the manuscript barely refers to metatranscriptomes and I wonder why that is. I think those results are important for underpinning the metabolic discussions especially in the context of adaptation to different springs with distinct env. conditions. Hence, I suggest to map the transcripts to the MAGs and also showing them independently from the MAGs as some of them might not be mapping.

Response: Indeed, what we did in our previous version of manuscript aligns precisely with the reviewer’s suggestion. We mapped metatranscriptomic data from four geothermal spring samples – one from the acidic DRTY-16 spring, two from neutral springs (JZ-2 and QQ pools), and one from the alkaline JZ-3 spring – onto all archaeal rMAGs. MAGs exhibiting highest expression among

archaea in the corresponding metatranscriptomic data were characterized, and genes were ranked based on their expression level (Fig. 5). We have done our best to discuss important results from transcriptional data within the context of major spring groups, particularly in the **Results** section related to sulfur metabolism (Lines 323 – 332), nitrogen metabolism (Lines 360 – 365, 375 – 386), and ecological roles of low-abundance lineages in the **Discussion** section (Lines 542 – 559). These findings are further elaborated in Figure 5, Supplementary Data 9, and Supplementary Data 11. We hope the reviewer recognizes that this is challenging at the scale of nearly 3,000 MAGs.

As suggested, we have made several modifications and additions related to metatranscriptomic data in the Revised Manuscript. This includes: 1) The updated Fig 5 differs slightly from the previous version. Initially, we depicted the expression at the order level by summing all expression values from individuals within the same order for a given gene (Fig 5e-j). However, in the latest version, we present expression data only for individual MAGs with high expression. The other three panels (Fig 5a-c) remain unchanged from the previous version. 2), we have improved our discussions on potential archaeal metabolisms by mentioning metatranscriptomic data in multiple places, including Lines 266, 307 – 308, 313 – 315, 370 – 372, 378, 479 – 481, and Figure 6. For instance, we now discuss the potential nitrogen cycle in diverse springs while considering the expression of genes involved in nitrogen cycling (Lines 370 – 372). 3) Moreover, in the carbon metabolism section, we have incorporated specific details on the abundance of transcripts encoding carbohydrate-active enzymes (CAZymes), complementing the content introduced in Lines 240 – 244 and supported by Supplementary Data 7.

Comment 2. This manuscript would benefit from co-occurrence analyses (e.g. which MAGs do co-occur?) in addition to gene network analyses (e.g. WGCNA). Currently, the paper is too much focusing on known pathways and their regulation. Thus, having a broader context will put those results into perspective and it will support some of the hypotheses (e.g., metabolic interaction between MAGs). For instance, those results are needed to provide evidence for the speculations between lines 377 and 407. Without evidence, this paragraph is too speculative and should therefore be deleted.

Response: We thank the reviewer for the valuable suggestions. In response, we have added co-occurrence network analyses in the Revised Manuscript (see Lines 723 – 732 in **Methods** for details). Unfortunately, gene network analyses, such as WGCNA, could not be performed due to the limited number of samples (only four). The co-occurrence networks were used to infer the putative metabolic and ecological relationships between some of the abundant Archaea. We showed that microbes belonging to *Sulfolobales* and *Thermoplasmata* were negatively correlated, which could be explained by their similar metabolisms. We show that temperature determines which order prevails over the other. Additionally, less abundant Archaea (e.g., small Archaea also known as DPANN) consistently appeared in the network, consistent with their known or suspected symbiotic relationships with other Archaea. We believe that the co-occurrence networks improved the focus and clarity of our narrative. Related descriptions have been added in the Revised Manuscript (Lines

Comment 3. I suggest to run a generalised linear model (GLM) with all environmental variables to build a relationship between the response (e.g. metabolic genes, MAGs) and predictors (e.g. env. variables) (e.g. lines 156-172). This will provide a more comprehensive approach for assessing which environmental variables are most important for the distinct metabolism identified in different springs. As many env. variables as possible should be included in this analyses.

Response: We appreciate the reviewer's insightful suggestions. The Supplementary Data 1 has been expanded to include additional physiochemical parameters related to Tengchong geothermal springs. Moreover, we applied a generalized linear model (GLM) to establish connections between species/functional diversity and environmental parameters using the *glm* function in the “stats” package (v. 4.2.2)⁷ in R software (<http://cran.r-project.org>). Statistical significance ($P < 0.05$) was assessed by conducting likelihood ratio tests (LRT) on the linear models, utilizing Type-II ANOVA in the “car” package (v. 3.0-10)⁸ in R.

The GLM analysis identified pH as the strongest correlate of both archaeal composition and metabolic functions ($\chi^2 = 40.8$, $P < 0.001$ for rMAGs; $\chi^2 = 46.0$, $P < 0.001$ for KOs) (Supplementary Table 1). Temperature was the second strongest correlate ($\chi^2 = 29.2$, $P < 0.001$ for rMAGs; $\chi^2 = 34.2$, $P < 0.001$ for KOs). Other variables, such as SO_4^{2-} and NO_2^- , were also significant ($P < 0.05$), which could guide the further investigation of biogeochemical cycles of sulfur and nitrogen in geothermal springs, as what we have detailed in the manuscript. These findings align with our results, highlighting that pH and temperature are critical factors influencing the archaeal community at species and function level. The related descriptions have been incorporated into the Revised Manuscript (Lines 168 – 175, 353 – 355, 717 – 721, Supplementary Table 1).

References:

1. Thompson, L. R. *et al.* A communal catalogue reveals Earth's multiscale microbial diversity. *Nature* **551**, 457–463 (2017).
2. Li, H. *et al.* The impact of temperature on microbial diversity and AOA activity in the Tengchong Geothermal Field, China. *Sci. Rep.* **5**, 17056 (2015).
3. Hou, W. *et al.* A Comprehensive Census of Microbial Diversity in Hot Springs of Tengchong, Yunnan Province China Using 16S rRNA Gene Pyrosequencing. *PLOS ONE* **8**, e53350 (2013).
4. Berg, I. A., Ramos-Vera, W. H., Petri, A., Huber, H. & Fuchs, G. Study of the distribution of autotrophic CO₂ fixation cycles in Crenarchaeota. *Microbiology* **156**, 256–269 (2010).
5. Friedman, J. & Alm, E. J. Inferring Correlation Networks from Genomic Survey Data. *PLOS Comput. Biol.* **8**, e1002687 (2012).
6. Bastian, M., Heymann, S. & Jacomy, M. Gephi: An Open Source Software for Exploring and Manipulating Networks. *Proc. Int. AAAI Conf. Web Soc. Media* **3**, 361–362 (2009).
7. Team, R. C., Team, M. R. C., Suggests, M. & Matrix, S. Package stats. *R Stats Package* (2018).
8. Fox, J. & Weisberg, S. *An R Companion to Applied Regression*. (SAGE Publications, 2011).

REVIEWER COMMENTS

Reviewer #1 (Remarks to the Author):

The manuscript has generally been well revised, but several essential issues still remain.

1. Biomass: The authors conducted quantitative PCR, but the data is not ecologically qualitative. The concentration of rRNA gene should be presented using units such as /ml or g hot spring water, sediment or biofilm. The concentration in the extracted DNA solution do not provide ecological information.
2. The authors conducted correlation analysis for archaeal lineage associated with the samples dominated by Archaea. I do not agree with their logic to figure out the ecological functions of the archaeal lineages. Complete analysis of bacterial MAGs as in the case of archaeal MAGs is not necessary, but the assembly of bacterial sequence and correlation analysis including bacterial groups in each microbial ecosystem must be required to understand the ecological function of archaeal lineages.
3. Operational direction of the central carbon metabolisms is sometimes influenced by energy metabolisms. However, the authors presented a section of carbon metabolism prior to that of the energy metabolism. The structure of the manuscript is not appropriate.

Specific comments

L118: Only Thermoproteia is presented as a phylum while other phylum names are not revealed here. It is bit confusing for readers.

L119 and through the manuscript: Nanobdellota (Göker M, Oren A. Valid publication of four additional phylum names. *Int J Syst Evol Microbiol.* 2023 Sep;73(9). doi: 10.1099/ijsem.0.006024.) instead of Nanoarchaeota has been validly published as well as lower taxonomic nomenclatures; <https://www.microbiologyresearch.org/content/journal/ijsem/10.1099/ijsem.0.005489>. I understand that the authors apply GTDB names, but the authors need to pay attention about the complicated situation about the nomenclature.

L137: defined instead of exist

L138: delete but

L146: Sulfolobales and Caldarchaeales belong to Thermoproteia. The order of order names should be reconsidered.

L183: I agree to use “extreme” in some of the cases in acidic hot springs, but not alkaline ones.

L196 and 347: Phylogenetic tree focusing on Sulfolobales is helpful for readers to understand the unique observation.

L209-: Did the authors find homologous energy conservation systems in Thermococcus?

See <https://journals.asm.org/doi/full/10.1128/mbio.02807-18>

L232-: As mentioned above, operational direction of acetate oxidation should be discussed more carefully.

L241: POR can operate CO₂ fixation under high CO₂ concentration (e.g. Steffens et al. 2021). The POR operation also contributes to acetate assimilation.

L318: inorganic carbon fixation

L325: Please clarify “nitrite correlated with archaeal community”.

L463: In my understanding, most of the acidic geothermal system are maintained by abiotic reaction, but not microbial process. Did the author find evidence other than genomic information such as very dense biomass of the archaeal lineage, stable isotopic signature etc?

Reviewer #2 (Remarks to the Author):

Happy with the revision. This is thorough work now with interesting insights based on a large dataset.

REVIEWER COMMENTS

Reviewer #1 (Remarks to the Author):

The manuscript has generally been well revised, but several essential issues still remain.

Response: We thank the reviewer for the positive assessment and valuable suggestions on our revision.

Comment 1. Biomass: The authors conducted quantitative PCR, but the data is not ecologically qualitative. The concentration of rRNA gene should be presented using units such as /ml or g hot spring water, sediment or biofilm. The concentration in the extracted DNA solution do not provide ecological information.

Response: Thank you for your suggestions. We extracted DNA from 1.5 g of hot spring sediments per sample. The unit has been modified, and the values have been adjusted accordingly, as presented in Table A below.

Sample	Microbial 16S rRNA gene (EMP-F, EMP-R)		Archaeal 16S rRNA gene (Arch519F, Arch915R)		Relative abundance of Archaea	
	Ct	Absolute copy number/ g	Ct	Absolute copy number/ g	qPCR	Metagenome
DRTY-2	23.9	3.67E+07	19.8	3.53E+07	96.4%	57.1 – 97.7% (ave. 83.1%)
DRTY-3	29.7	8.90E+05	25.3	7.00E+05	78.7%	56.4 – 87.1% (ave. 72.5%)
DRTY-9	25.5	1.31E+07	22.0	7.53E+06	57.4%	74.7 – 97.8% (ave. 86.9%)
DRTY-19	24.7	2.28E+07	21.6	1.01E+07	44.3%	34.5 – 56.9% (ave. 45.7%)
JZ-1	26.2	3.66E+08	26.2	2.85E+08	77.9%	10.1 – 58.3% (ave. 36.7%)
JZ-3	19.6	6.00E+08	18.7	7.83E+07	13.1%	0.1 – 6.9% (ave. 2.9%)
JZ-4	21.6	1.67E+08	20.6	2.05E+07	12.3%	0.2 – 7.4% (ave. 2.7%)
QQ	21.9	1.39E+08	19.7	3.67E+07	26.4%	3.1 – 20.1% (ave. 9.6%)

Table A. Absolute and relative abundance of Archaea determined by qPCR experiments. Samples were collected from Tengchong geothermal springs in Aug, 2023.

Comment 2. The authors conducted correlation analysis for archaeal lineage associated with the samples dominated by Archaea. I do not agree with their logic to figure out the ecological functions of the archaeal lineages. Complete analysis of bacterial MAGs as in the case of archaeal MAGs is not necessary, but the assembly of bacterial sequence and correlation analysis including bacterial groups in each microbial ecosystem must be required to understand the ecological function of archaeal lineages.

Response: In response to the suggestion, we conducted co-occurrence network reconstruction with the inclusion of Bacteria. Briefly, we scanned both bacterial and archaeal *rpS3* genes in all assembled scaffolds of sampling sites dominated by Archaea. The obtained *rpS3* genes were used to build an OTU table by calculating the relative abundances of each *rpS3* in each sample. Subsequently, the same procedures were performed to build co-occurrence network (see Methods in the Revised Manuscript). As illustrated in the Figure A below, integrating Bacteria into the

network did not significantly alter the correlations among Archaea. The relationships among archaeal lineages observed in the microbial *rpS3* networks align closely with those in archaeal MAG networks (Figure 6b-c in the manuscript). For example, consistent negative correlations are still observed between the dominant lineages *Thermoplasmatota* and *Sulfolobales* in thermoacidic springs, while positive correlations persist between abundant *Thermoproteales* and *Sulfolobales* in thermoalkaline springs. Interestingly, given the predominant abundance of Archaea in these polyextreme springs (thermoacidic: average 62 ± 11 °C, pH = 3.4 ± 1.0 ; thermoalkaline: average 84 ± 14 °C, pH = 9.1 ± 0.4), only a few positive links between Bacteria and Archaea were observed. Most links between Archaea and Bacteria exhibit negative correlations, suggesting possible competition. In thermoacidic springs, only 8.9% (36/404) of the correlations were positive, and the majority of those linked *Thermotogota* with Archaea (Table B). Similarly, in thermoalkaline springs, only 6.7% (12/178) of positive links were identified between Bacteria and Archaea (Table C). Collectively, our results indicate that the general pattern among Archaea remained consistent even when considering Bacteria and Bacteria may have minimal impact on the growth of Archaea. Given that Bacteria are not the focus in this study, conducting relevant analyses without their consideration might be preferable.

Figure A. Co-occurrence networks between Archaea and Bacteria based on *rpS3* protein sequences in thermoacidic (a) and thermoalkaline (b) springs. Some archaeal *rpS3* sequences were only identified at the phylum level and represented in circles with alphabetic labels.

Archaea		Bacteria	Campylobacterota	Thermotogota		
			NODE_228	NODE_230765	NODE_25110	NODE_257141
Micrarchaeales	NODE_1036		1	1	1	1
	NODE_1123129			1	1	1
	NODE_19202		1	1	1	1
	NODE_36634					1
	NODE_368		1	1	1	1
Thermoplasmatota	NODE_451			1	1	1
	NODE_102570			1	1	1
	NODE_119286			1		1
	NODE_158289			1	1	1
Thermoproteota	NODE_529			1	1	1
	NODE_148051		1	1	1	1
	NODE_743				1	1

Table B. Positive correlations between Archaea and Bacteria based on *rpS3* protein sequences in thermoacidic springs

Archaea		Bacteria	Aquificota					Armatimonadota		Bipolaricaulota
			NODE_11773	NODE_13743	NODE_33850	NODE_379951	NODE_6410	NODE_1098	NODE_754	NODE_69021
Sulfolobales	NODE_16896						1			
	NODE_54272							1	1	1
	NODE_61396								1	1
	NODE_6843								1	1
Thermoproteales	NODE_74860	1	1	1	1					

Table C. Positive correlations between Archaea and Bacteria based on *rpS3* protein sequences in thermoalkaline springs

Comment 3. Operational direction of the central carbon metabolisms is sometimes influenced by energy metabolisms. However, the authors presented a section of carbon metabolism prior to that of the energy metabolism. The structure of the manuscript is not appropriate.

Response: Per reviewer's request, we have restructured the manuscript to present the carbon fixation and methane metabolism first (Lines 212 – 232), followed by acetate metabolism including both consumption and production processes (Lines 253 – 278). Finally, we have presented other heterotrophic metabolisms, such as carbohydrate degradation (Lines 279 – 304).

Specific comments

Comment 4. L118: Only Thermoproteia is presented as a phylum while other phylum names are not revealed here. It is bit confusing for readers.

Response: We need to clarify that all taxa presented here are at class level. *Thermoproteia* is a class within the phylum *Thermoproteota*. *Nitrososphaeria*, *Bathyarchaeia*, *Micrarchaeia*, and *Nanoarchaeia* are also class names. To avoid confusion, we have removed mention of the phylum *Thermoproteota*, but all class names have been retained (Line 118).

Comment 5. L119 and through the manuscript: Nanobdellota (Göker M, Oren A. Valid publication of four additional phylum names. *Int J Syst Evol Microbiol*. 2023 Sep;73(9). doi: 10.1099/ijsem.0.006024.) instead of Nanoarchaeota has been validly published as well as lower taxonomic nomenclatures; <https://www.microbiologyresearch.org/content/journal/ijsem/10.1099/ijsem.0.005489>. I understand

that the authors apply GTDB names, but the authors need to pay attention about the complicated situation about the nomenclature.

Response: We appreciate the reviewer's recommendation to update the taxonomy. While we have chosen to use the names in the most recent GTDB release for consistency, we have added clarification in our paper that this new name (along with names for parent and daughter taxa) has been validly published and cited the effective publication (Lines 119 – 120, 125, 145, 152, 419, 522). We have maintained the GTDB nomenclature in the figures.

Comment 6. L137: defined instead of exist

Response: Done as suggested (Line 138).

Comment 7. L138: delete but

Response: Done as suggested (Line 138).

Comment 8. L146: Sulfolobales and Caldarchaeales belong to Thermoproteia. The order of order names should be reconsidered.

Response: Done as suggested (Lines 146 – 148).

Comment 9. L183: I agree to use “extreme” in some of the cases in acidic hot springs, but not alkaline ones.

Response: We acknowledge that alkaline springs may not universally be considered extreme environments. However, it's worth noting that the majority of alkaline hot springs in our study are indeed hyperthermal alkaline springs (temperatures $\geq 80^{\circ}\text{C}$ and $\text{pH} \geq 8.5$) and that these springs are also dominated by Archaea. Therefore, we believe it's appropriate to use "extreme" in this context.

Comment 10. L196 and 347: Phylogenetic tree focusing on Sulfolobales is helpful for readers to understand the unique observation.

Response: Done as suggested. We have included a phylogenomic tree of the *Sulfolobales* in Supplementary Figure 6 (Lines 196 – 200, 400 – 401). Furthermore, the family names within *Sulfolobales* enriched in acidic and alkaline springs have been specified.

Comment 11. L209-: Did the authors find homologous energy conservation systems in *Thermococcus*? See <https://journals.asm.org/doi/full/10.1128/mbio.02807-18>

Response: *Thermococcus* were not detected in our data set. We assume the reviewer is inquiring about whether our archaeal genomes contain homologous ferredoxin (Fd) proteins (Fd-1: TK1694, COG1141; Fd-2: TK1087, COG1149; Fd-3: TK2012, COG1146) as mentioned in the mBio paper. As each Fd homolog in *Thermococcus* is associated with a COG number, we proceeded to search for all genes within our archaeal rMAGs against the COG database¹ using DIAMOND (v.2.0.11.149)² with an E value of $< 1\text{e-}5$ and hit score ≥ 100 . Results show that Fd homologs were

successfully identified in 112 out of 603 archaeal rMAGs (see Table D below). Fd-1 and Fd-3 were more commonly detected, whereas none of the rMAGs contain Fd-2 homologs. Multiple orders within the phylum *Thermoproteota* harbored Fd-1, which may primarily facilitate the transfer of electrons from glucose and/or amino acids to oxidoreductases, which reduce NAD(P)^+ to NAD(P)H^3 . Regarding Fd-3, it was detected not only in the phylum *Thermoproteota* but also in the phyla *Aenigmataarchaeota*, B1Sed10-29, *Micrarchaeota*, *Nanoarchaeota*, *Halobacteriota*, and *Thermoplasmatota*. Fd-3 primarily participates in hydrogen production by facilitating electron transfer to hydrogen-evolving membrane-bound hydrogenases, particularly Group 4 [NiFe]-hydrogenases, which were found to be diverse and widely distributed among rMAGs reconstructed from Tengchong springs. Corresponding results about ferredoxins Fd-1 and Fd-3 were added to the Revised Manuscript (Lines 267 – 269 for Fd-1 and Lines 448 – 449 for Fd-3).

Lineages	Only Fd-1 (TK1694)	Only Fd-3 (TK2012)	Both Fd-1 and Fd-3
p__Aenigmataarchaeota;c__Aenigmataarchaeia;o__GW2011-AR5	-	1	-
p__B1Sed10-29;c__B1Sed10-29;o__B1Sed10-29	-	1	-
p__Halobacteriota;c__Halobacteria;o__Halobacteriales	1	-	-
p__Halobacteriota;c__Methanosarcinia;o__Methanotriconales	-	1	-
p__Methanobacteriota;c__Methanobacteria;o__Methanobacteriales	1	-	-
p__Micrarchaeota;c__Micrarchaeia;o__Anstonellales	-	1	-
p__Micrarchaeota;c__Micrarchaeia;o__CAILAH01	-	1	-
p__Nanoarchaeota;c__Nanoarchaeia;o__Pacearchaeales	-	6	-
p__Nanoarchaeota;c__Nanoarchaeia;o__SCGC-AAA011-G17	-	1	-
p__Thermoplasmatota;c__Thermoplasmatata;o__ARK-15	-	1	-
p__Thermoplasmatota;c__Thermoplasmatata;o__DTKX01	1	-	-
p__Thermoplasmatota;c__Thermoplasmatata;o__Thermoplasmatatales	-	5	-
p__Thermoproteota;c__Bathyarchaeia;o__B24	-	1	-
p__Thermoproteota;c__Bathyarchaeia;o__B26-1	4	16	-
p__Thermoproteota;c__Korarchaeia;o__Korarchaeales	3	-	1
p__Thermoproteota;c__Methanomethylia;o__Nezhaarchaeales	1	-	-
p__Thermoproteota;c__Nitrososphaeria;o__Conexivisphaerales	-	1	-
p__Thermoproteota;c__Nitrososphaeria;o__JACIWG01	3	1	1
p__Thermoproteota;c__Nitrososphaeria;o__Nitrososphaerales	3	-	1
p__Thermoproteota;c__Thermoproteia;o__Sulfolobales	33	1	6
p__Thermoproteota;c__Thermoproteia;o__Thermofilales	8	-	-
p__Thermoproteota;c__Thermoproteia;o__Thermoproteales	7	-	-

Table D. The occurrence frequency of ferredoxin proteins in archaeal rMAGs reconstructed from Tengchong geothermal springs

Comment 12. L232-: As mentioned above, operational direction of acetate oxidation should be discussed more carefully.

Response: Thanks for your reminder regarding the operational direction of acetate oxidation. Based on the kinetic parameters, the AMP-forming acetyl-CoA synthetases (ACS) catalyze the conversion of acetate to acetyl-CoA, exhibiting a lower apparent K_m value for acetate, indicating higher affinity⁴. Conversely, the ADP-forming acetyl-CoA synthetase (ACD) demonstrates higher affinity for acetyl-CoA, with a much lower apparent K_m value compared to acetate⁵⁻⁷. While the reaction mediated by ACD is reversible, in situations with limited carbon sources or when acetate is the sole carbon source, this enzyme can undergo the reverse reaction facilitating acetate assimilation, in conjunction with the POR enzyme converting acetyl-CoA to pyruvate^{7,8}. We have already incorporated the occurrence of the reverse reaction of ACD in the revised manuscript (Lines 271 –

273).

Comment 13. L241: POR can operate CO₂ fixation under high CO₂ concentration (e.g. Steffens et al. 2021). The POR operation also contributes to acetate assimilation.

Response: We acknowledge that POR can participate in several pathways. Since no known archaea are capable to fix carbon via the rTCA or roTCA cycle, as emphasized in paper by Steffens et al. (2021), we have omitted discussion of this in the Revised Manuscript. However, we agree with the reviewer that POR may contribute to acetate assimilation. This information has been added in the Revised Manuscript (Lines 271 – 273). Additionally, we have provided details on the conditions conducive to the occurrence of this reaction. In environments with limited carbon sources, some Archaea can utilize reverse acetate-CoA ligases (ACD) to assimilate acetate to acetyl-CoA, which is then converted to pyruvate mediated by POR⁸. Additionally, it's worth noting that most Archaea encoding ACD also encode the *por* gene.

Comment 14. L318: inorganic carbon fixation

Response: Done as suggested (Line 371).

Comment 15. L325: Please clarify “nitrite correlated with archaeal community”.

Response: Done as suggested (Lines 377 – 379), and detailed statistics are shown in lines 169 – 170 of the manuscript.

Comment 16. L463: In my understanding, most of the acidic geothermal system are maintained by abiotic reaction, but not microbial process. Did the author find evidence other than genomic information such as very dense biomass of the archaeal lineage, stable isotopic signature etc?

Response: We deleted “microbially generated” to be conservative on the source of sulfide (Lines 519 – 520), but we believe microbial sulfide oxidation to be the major source of sulfuric acid. Kinetics experiments with sulfide in both freshwater and seawater have shown that sulfide oxidation is extremely slow below pH 6, where H₂S dominates^{9,10}. Also, experiments with mats collected from acidic springs in Yellowstone National Park have shown that heat-killed mats consumed hydrogen sulfide about an order of magnitude slower than untreated mats¹¹. We have also conducted experiments in which hydrothermal water was either filtered through 0.2-micron filters or not filtered and then incubated in the field in HDPE bottles and the rates of sulfide oxidation were >10-fold higher in unfiltered water samples compared to filtered water samples. Those data were never published partly because filters could also remove suspended minerals that could act as catalysts, although the spring water was very clear. A counter to the biological oxidation of sulfide in Yellowstone springs based on lab vs field sulfide oxidation experiments has also been published¹².

Reviewer #2 (Remarks to the Author):

Happy with the revision. This is thorough work now with interesting insights based on a large dataset.

Response: We genuinely appreciate the recognition from the reviewer!

References

1. Galperin, M. Y. *et al.* COG database update: focus on microbial diversity, model organisms, and widespread pathogens. *Nucleic Acids Res.* **49**, D274–D281 (2021).
2. Buchfink, B., Reuter, K. & Drost, H.-G. Sensitive protein alignments at tree-of-life scale using DIAMOND. *Nat. Methods* **18**, 366–368 (2021).
3. Burkhart, B. W., Febvre, H. P. & Santangelo, T. J. Distinct Physiological Roles of the Three Ferredoxins Encoded in the Hyperthermophilic Archaeon *Thermococcus kodakarensis*. *mBio* **10**, 10.1128/mbio.02807-18 (2019).
4. Kuprat, T., Ortjohann, M., Johnsen, U. & Schönheit, P. Glucose Metabolism and Acetate Switch in Archaea: the Enzymes in *Haloferax volcanii*. *J. Bacteriol.* **203**, 10.1128/jb.00690-20 (2021).
5. Glasemacher, J., Bock, A.-K., Schmid, R. & Schönheit, P. Purification and Properties of Acetyl-CoA Synthetase (ADP-forming), an Archaeal Enzyme of Acetate Formation and ATP Synthesis, from the Hyperthermophile *Pyrococcus furiosus*. *Eur. J. Biochem.* **244**, 561–567 (1997).
6. Musfeldt, M., Selig, M. & Schönheit, P. Acetyl Coenzyme A Synthetase (ADP Forming) from the Hyperthermophilic Archaeon *Pyrococcus furiosus*: Identification, Cloning, Separate Expression of the Encoding Genes, *acdAI* and *acdBI*, in *Escherichia coli*, and In Vitro Reconstitution of the Active Heterotetrameric Enzyme from Its Recombinant Subunits. *J. Bacteriol.* **181**, 5885–5888 (1999).
7. Bräsen, C., Schmidt, M., Grötzinger, J. & Schönheit, P. Reaction Mechanism and Structural Model of ADP-forming Acetyl-CoA Synthetase from the Hyperthermophilic Archaeon *Pyrococcus furiosus*. *J. Biol. Chem.* **283**, 15409–15418 (2008).
8. Sorokin, D. Y. *et al.* Elemental sulfur and acetate can support life of a novel strictly anaerobic haloarchaeon. *ISME J.* **10**, 240–252 (2016).
9. Kennedy, C. & Morris, J. C. Kinetics of oxidation of aqueous sulfide by O₂. *Environ. Sci. Technol.* **6**, 529–537 (1972).
10. Zhang, J.-Z. & Millero, F. The products of oxidation of H₂S in seawater. *Geochim. Cosmochim. Acta - GEOCHIM COSMOCHIM ACTA* **57**, 1705–1718 (1993).
11. D'Imperio, S. *et al.* Relative Importance of H₂ and H₂S as Energy Sources for Primary Production in Geothermal Springs. *Appl. Environ. Microbiol.* **74**, 5802–5808 (2008).
12. Schoen, R. Rate of Sulfuric Acid Formation in Yellowstone National Park. *GSA Bull.* **80**, 643–650 (1969).

REVIEWERS' COMMENTS

Reviewer #1 (Remarks to the Author):

I sincerely appreciate the efforts of the authors, but still found issues as listed below.

Major issues

Microbial abundance (qPCR data) should be integrated into Supplementary Table 1, and statistics. Using the data set, the authors can obtain semi-quantitative data set to figure out and discuss the microbial ecosystem more clearly by a combination with the quantitative data and qualitative data such as the relative abundance in the metagenomic libraries (e.g. Fig S4, S5 etc).

L453-

The result of the network analysis including bacterial population is the only scientific justification why the authors focus on archaeal population in the microbial ecosystems. Thus, the result should be presented as a main figure, or at least a part of supplementary materials.

Nomenclature

If the authors prefer to use GTDB nomenclature but not others, it is better to declare the preference in the introduction section or the head of the result section. This would minimize the confusing in nomenclature. More comments are also listed in minor comments.

Minor comments

L119: validated instead of renamed

L125: What does Nitrosospharia_A?

L148, 525: (within Aenigmataarchaeia)

L152, 524: (within Micrarchaeia)

L152: Nanobdelalles instead of Nanobdellota

Fig S9: The names of superphyla are suddenly (probably only) appeared here. Note that the names of superphyla have been validated as Kingdoms very recently in IJSEM (doi.org/10.1099/ijsem.0.006242).

REVIEWER COMMENTS

Reviewer #1 (Remarks to the Author):

I sincerely appreciate the efforts of the authors, but still found issues as listed below.

Response: Thank you very much for your serious, helpful suggestions and positive affirmation of our article!

Major issues

Microbial abundance (qPCR data) should be integrated into Supplementary Table 1, and statistics. Using the data set, the authors can obtain semi-quantitative data set to figure out and discuss the microbial ecosystem more clearly by a combination with the quantitative data and qualitative data such as the relative abundance in the metagenomic libraries (e.g. Fig S4, S5 etc).

Response: Per reviewer's request, we have incorporated the qPCR data into Supplementary Table 1, and the corresponding results have been described in the Revised Manuscript (Lines 117 – 118 and 194 – 195).

L453-

The result of the network analysis including bacterial population is the only scientific justification why the authors focus on archaeal population in the microbial ecosystems. Thus, the result should be presented as a main figure, or at least a part of supplementary materials.

Response: As suggested, we have presented microbial networks (encompassing both Archaea and Bacteria) on FigShare repository (refer to “Microbial networks” at <https://doi.org/10.6084/m9.figshare.25650441>). These results have been incorporated into the Revised Manuscript (Lines 418 – 424).

Nomenclature

If the authors prefer to use GTDB nomenclature but not others, it is better to declare the preference in the introduction section or the head of the result section. This would minimize the confusing in nomenclature. More comments are also listed in minor comments.

Response: Following the suggestion, we have included a declaration regarding the use of GTDB nomenclature in the Revised Manuscript (Lines 98 and 109).

Minor comments

L119: validated instead of renamed

Response: Done as suggested (Line 122).

L125: What does Nitrosospharia_A?

Response: It is the class name of order *Caldarchaeales* (previously known as *Aigarchaeota*) as defined by GTDB (https://gtdb.ecogenomic.org/tree?r=c_Nitrososphaeria_A), which exhibits

close relationship with *Nitrososphaeria*.

L148, 525: (within Aenigmatarchaeia)

Response: Done as suggested (Lines 149 and 489).

L152, 524: (within Micrarchaeia)

Response: Done as suggested (Lines 154 and 489).

L152: Nanobdelalles instead of Nanobdellota

Response: Done as suggested (Lines 154)

Fig S9: The names of superphyla are suddenly (probably only) appeared here. Note that the names of superphyla have been validated as Kingdoms very recently in IJSEM (doi.org/10.1099/ijsem.0.006242).

Response: Revised as recommended (refer to Supplementary Fig. 9).